# Elucidating the assembly of gas vesicles by systematic protein-protein interaction analysis

Manuel Iburg [1,3], Andrew P Anderson [1,3], Vivian T Wong [1], Erica D Anton [1], Art He [1] & George J Lu [1,2 ✉]

## Abstract

**Gas vesicles (GVs) are gas-filled microbial organelles formed by unique 3-nm thick, amphipathic, force-bearing protein shells, which can withstand multiple atmospheric pressures and maintain a physically stable air bubble with megapascal surface tension. However, the molecular process of GV assembly remains elusive. To begin understanding this process, we have devised a high-throughput in vivo assay to determine the interactions of all 11 proteins in the pNL29 GV operon. Complete or partial deletions of the operon establish interdependent relationships among GV proteins during assembly. We also examine the tolerance of the GV assembly process to protein mutations and the cellular burdens caused by GV proteins. Clusters of GV protein interactions are revealed, proposing plausible protein complexes that are important for GV assembly. We anticipate our findings will set the stage for designing GVs that efficiently assemble in heterologous hosts during biomedical applications.**

**Keywords** gas vesicles; microbial organelle; acoustic reporter gene; macromolecular assembly; protein-protein interaction
**Subject Categories** Biotechnology & Synthetic Biology; Microbiology, Virology & Host Pathogen Interaction

## Introduction

Gas vesicles (GVs) are a class of gas-filled protein organelles evolved in photosynthetic microbes, which use them as flotation devices to compete for the surface of the water and maximize photosynthesis (Pfeifer, 2012; Walsby, 1994). Besides this native function, the interest in GVs has grown substantially due to the recent development of a wide range of biomedical applications based on these genetically encodable nanostructures, including gene expression imaging by ultrasound, MRI, and optical methods (Bourdeau et al, 2018; Farhadi et al, 2019; Hurt et al, 2023; Lu et al, 2020; Lu et al, 2018; Sawyer et al, 2021; Shapiro et al, 2014a; Shapiro et al, 2014b), protease sensing (Lakshmanan et al, 2020), payload

delivery (Bar-Zion et al, 2021), cellular manipulation (Wu et al, 2023; Yang et al, 2023), gas delivery (Fernando and Gariepy, 2020; Song et al, 2020), cell tracking (Hao et al, 2023), and pressure sensing (Anthis et al, 2022). This rapidly growing set of applications has now demanded a deeper understanding of the biology of GV formation to facilitate their molecular engineering.

As a product of evolution and natural selection, GVs have a fundamentally different design compared to man-made synthetic bubbles. Bubbles usually trap air inside at a non-equilibrium state, leading to a finite lifetime, and moreover, as the diameter of bubbles reduces, a higher Laplace pressure builds up inside, leading to a faster dissipation of entrapped air (Brennen, 1995). For GVs, however, the protein shell is permeable to individual molecules of both gas and water, and GVs maintain an inner gas compartment by having a hydrophobic inner surface that prevents heterogeneous condensation of water molecules into liquid. Meanwhile, the small size of the gas compartments minimizes the chance of homogeneous condensation (Walsby et al, 1992). In combination with a rigid shell that prevents the shrinking of the gas compartments, GVs outcompete the best synthetic bubbles in both the lifetime and the size—nanobubbles have only recently reached <200 nm in diameter and a lifetime of minutes (Counil et al, 2022), while GVs can be made with diameters smaller than 100 nm and stable for months (Ling et al, 2023; Shen et al, 2023).

This unique design principle of GVs necessitates a set of complex cellular machinery to support their assembly. For example, the major shell proteins have highly hydrophobic inner surfaces and aggregate even in the presence of strong detergents such as sodium dodecyl sulfate (SDS) (Walsby and Hayes, 1988). It is thus likely that the shell proteins are coordinated constantly by other proteins from their synthesis at the ribosomes until their insertion into the GV nanostructures. In addition, different from synthetic bubbles to which gas is exogenously provided, the gas inside GVs is equilibrated from that dissolved in the surrounding fluid, which hints at an interesting initiation process and the subsequent elongation of the gas compartment. Notably, this is in contrast to most of the other protein nanostructures described to date, which are formed by shell proteins with a propensity of self-assembly (Bale et al, 2016; Luo et al, 2016; Sutter et al, 2008; Sutter et al, 2017; Suzuki et al, 2016).

Despite the recognition of the unique assembly mechanism, the understanding of the functional roles of the gas vesicle proteins (Gvps)

[1]Department of Bioengineering, Rice University, Houston, TX 77005, USA. [2]Department of BioSciences, Rice University, Houston, TX 77005, USA. [3]These authors contributed equally: Manuel Iburg, Andrew P Anderson. ✉E-mail: george.lu@rice.edu

and the molecular assembly process has severely lagged behind. Only the major shell protein, which is usually denoted as *gvpA*, and the minor shell protein, *gvpC*, have clearly defined functions (Dutka et al, 2023; Huber et al, 2023). The remaining genes often have undefined functions in a given GV operon, which usually contains on the order of 10 genes (Hurt et al, 2023; Pfeifer, 2012). Taking one of the most well-studied GV operons, the 9-gene *gvpACNJKGFVW* from *Anabaena flos-aquae* as an example, the seven genes after *gvpA* and *gvpC* were believed to be assembly factors or minor constituents of GVs. Among them, GvpF is the only one with high-resolution crystal structures determined from orthologs, but neither structure provided an explanation of how the protein may function in assembling GVs (Cai et al, 2020; Xu et al, 2014). Much less is known for the other assembly proteins, and it is also puzzling to see the existence of this 7-gene cassette is contradicting the minimal eight essential genes determined in the case of the pNL29 GV operon originally cloned from *Bacillus megaterium* (Farhadi et al, 2019).

Recently, a number of studies have investigated subsets of the protein–protein interactions in the GV operon of *H. salinarum*, utilizing heterologous expression in *H. volcanii* or in vivo assays to monitor 1-on-1 protein interactions (Jost and Pfeifer, 2022; Tavlaridou et al, 2014; Völkner et al, 2020; Winter et al, 2018). In this study, we build on this example to make one of the first steps toward understanding the GV assembly mechanism by screening all possible protein–protein interactions in the GV operon derived from *B. megaterium*, which is commonly used for expression in *E. coli*. We will add an additional layer to previous studies by screening in the presence of the other GV proteins, essentially measuring all interactions as GVs are being assembled and subsequently subtracting proteins from the operon to elucidate interdependent interactions.

## Results

### Streamlining the Gvp interaction screen minimizes cloning workload and maximizes information

We decided to focus our investigation on the pNL29 operon originally cloned from *B. megaterium* (Li and Cannon, 1998) (Fig. 1A–C) for three reasons: (i) many recent biomedical applications, including both bacterial and mammalian acoustic reporter genes (Bourdeau et al, 2018; Farhadi et al, 2019) use the assembly factor proteins of the pNL29 operon, and thus the knowledge of these proteins will directly contribute to the engineering and design of GVs; (ii) the high-resolution structure of the GVs encoded by this operon was recently determined (Huber et al, 2023), opening the door to potential structure-based modeling of the assembly process using these structures; and (iii) the pNL29 operon allows GVs to be heterologously expressed and assembled in *E. coli* (Lakshmanan et al, 2017), which provides a robust platform to leverage genetic tools to study the assembly mechanism. The *B. megaterium*-derived GV operon contains the Gvps A2RNFGLSKJTU and notably, the second GvpA homolog of *B. megaterium* has been referred to as GvpB (Li and Cannon, 1998), but is now recognized as GvpA2 (Huber et al, 2023) and will be labeled as such in the following.

Following this choice, we considered the number of conditions we would need to probe (Fig. 1D). The pNL29 operon contains 11 genes, and thus there is a total of 55 potential heterologous

protein–protein interactions and another 11 self-interactions. We chose a NanoLuc-based split-luciferase complementation assay ("NanoBiT") to determine their interactions because of the high dynamic range, low background, commercially available substrates, and compatibility with experiments in living cells (Cabantous et al, 2005; Dixon et al, 2016). For each protein–protein interaction, this method requires us to consider all possible geometries of fusion proteins and include an option to supplement none, all, or some of the other Gvps. We concluded that 968 or more conditions would have to be tested (see also Appendix Fig. S5B). To minimize the molecular cloning workload of this comprehensive assay, we created three plasmid groups (I, II, III) with independent origins of replication and resistance markers that could be co-transformed into *E. coli* in parallel (Fig. 1E; Appendix Fig. S1) which is further described in the methods section. Combining any plasmids with interacting Gvps fused to the NanoBit split-luciferase would now lead to complementation and luciferase signal (Fig. 1F). Overall, this strategy successfully reduced the number of cloned plasmids to 128 (Table EV1) and created a modular plasmid framework to mix-and-match all Gvp interactions we would need to screen in vivo.

We then continued to establish an experimental workflow to maximize the information gained per protein–protein interaction assay performed. We probed protein–protein interactions in triplicates at two different timepoints (as described in the next section), and for a subset of the samples, we confirmed the expression of gas vesicles by phase-contrast microscopy and the expression of fusion proteins by western blot. We also monitored the transformation efficiency of our three-plasmid transformation, and the significance of these additional datapoints will be discussed in the sections "A fraction of Gvp fusion proteins enable GV formation but affect cell viability" and "All GV fusion proteins are expressed, but vary in stability" (Fig. 2A–G).

Finally, we performed our protein–protein interaction assays in a defined order to minimize redundancy. The appendix (Appendix Results section) contains a more in-depth description of our considerations for designing plasmids, the number of conditions to be assayed, considerations for focusing solely on in vivo protein–protein interaction analysis and our experimental workflow.

### Broad screening of the GV operon reveals a dense protein interaction network

With the abovementioned experimental workflow, we ended up testing a total of 1008 protein–protein interactions, each with $N = 3$ biological replicates. Each experiment included a buffer blank, a positive control that assayed rapamycin-induced FKBP12-FRB dimerization and a negative control of FKBP12 or FRB fusion protein with a GV fusion protein. To quantify the protein–protein interactions, we defined the rapamycin-inducible positive control in each experiment as 100% strength and scored signals as a fraction of the control. We considered the FRB-FKBP12 interaction in the presence of rapamycin as an inducible, physiologically relevant, high-affinity interaction (Banaszynski et al, 2005) and thus, a benchmark of protein interactions.

While a stronger signal from luciferase complementation does not necessarily indicate a more physiologically relevant protein–protein interaction, we decided to bin our results to help readers decode the bulk of results reported. Interactions >20% were defined as high, 10–20% as medium, and 5–10% as low. The bins

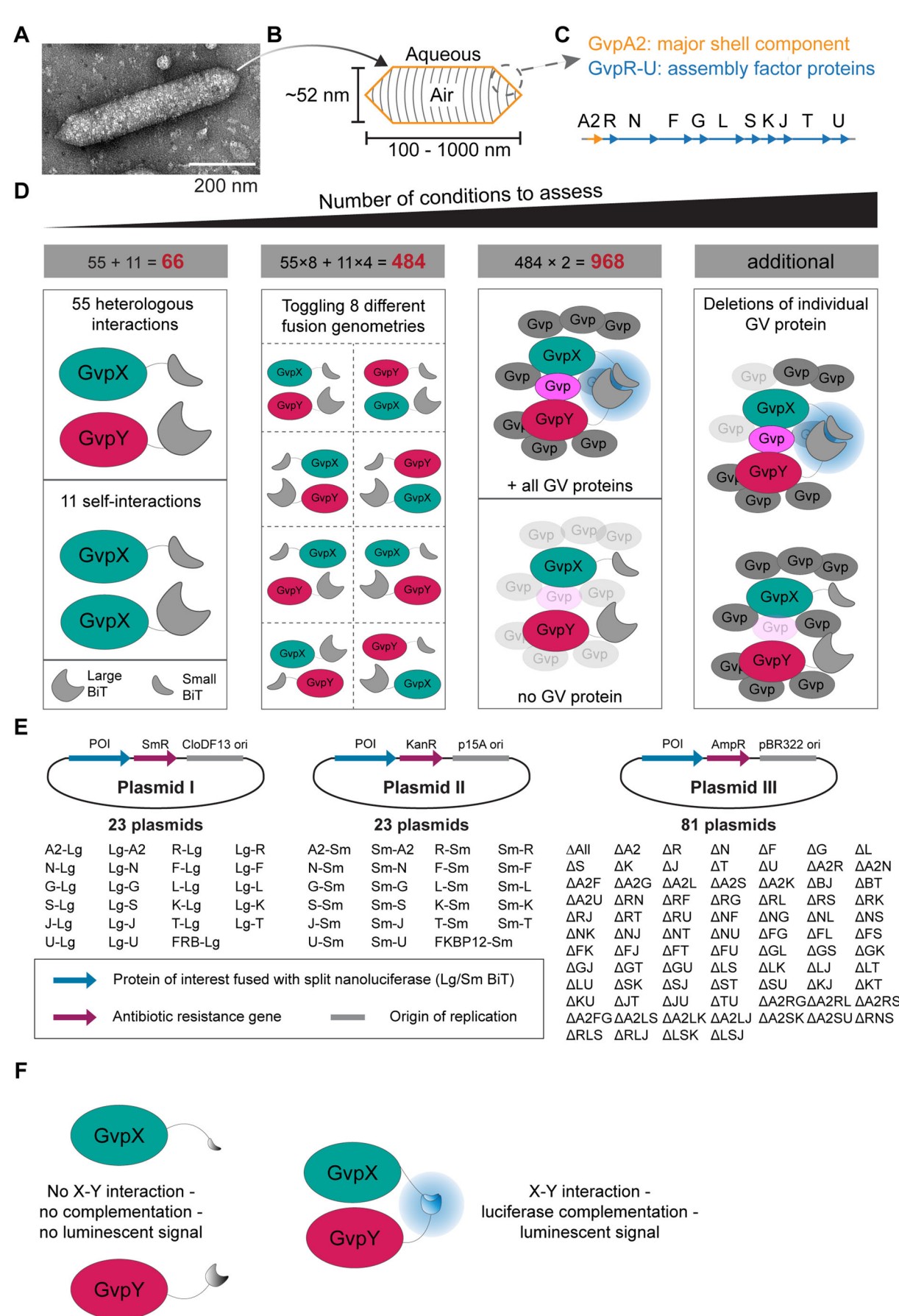

◄ **Figure 1. A mix-and-match plasmid library to assay protein interactions in the GV operon.**

(A, B) A transmission electron microscopy (TEM) image and a schematic representation of GVs encoded by the pNL29 operon. The scale bar represents 200 nm. (C) The operon consists of the structural component (labeled in yellow) and assembly factors (labeled in blue). (D) Calculating the numbers of genetic conditions through the 4 layers of variables. GV proteins fused to Large BiT and Small BiT of the split NanoLuc are marked in mint-green and red, respectively. In the 1st layer, the heterologous and self-interactions of all 11 GV proteins result in 66 conditions, and in the 2nd layer, toggling the fusion geometries brings the number to a total of 484 conditions excluding controls. In the 3rd layer, to determine if protein–protein interaction is direct or interdependent on a third GV protein (marked in magenta), all interactions will be measured in the presence and absence of the other GV proteins (knocked-out GV proteins marked in translucent gray). If the interaction is dependent on a third protein, the bioluminescence signal (marked in green) will only show up when all GV proteins are present. In the 4th layer, to search for the identity of the third GV protein (marked in magenta), partial knockout of GV proteins will be employed, which further expands the number of conditions. (E) Minimizing the number of plasmids needed to construct by the strategy of mix-and-match of three plasmids. The origins of replication, resistance markers, and proteins of interest (POI) are marked in gray, magenta, and blue, respectively. The abbreviated names of all 127 plasmids are listed. Lg: Large BiT. Sm: Small BiT. A2, R, N, F, G, L, S, K, J, T, U: individual GV protein, for example, A2 stands for GvpA2. Δ stands for a plasmid that contains all 11 GV proteins except for those indicated. See also the plasmid map in Appendix Fig. S1 and the full plasmid list in Table EV1. (F) A sketch illustrating the luciferase complementation assay. Left: two fusion proteins X and Y do not interact. The fused split-luciferase reporters cannot self-complement and no luciferase signal is observed. Right: two fusion proteins X and Y interact. The fused split-luciferase moieties complement each other due to the physical proximity of X and Y and luciferase signal is observed. Source data are available online for this figure.

were chosen after data acquisition, creating three groups of roughly equal population, and serve to visually decode the data, but do not relate to biological relevance or statistical significance of contained datapoints. Following the maxim of underreporting rather than overreporting protein interactions, we did not count signals with low statistical significance as protein–protein interactions. Signals <5% of the positive control were excluded from our final analysis to minimize the chance of including noise-level measurements in our data. In this way, 24 interactions were grouped as "high", 72 interactions were grouped as "medium", and 38 interactions were grouped as "low", resulting in 134 out of 1008, or ~10% of all measurements being counted as protein–protein interactions within the operon. Redundant interactions were observed between two GV proteins of different configurations of fusion, and we consolidated these interactions by taking the strongest one as the representative interaction, which resulted in a final set of 13 high, 5 medium, and 9 low interactions.

Approximately half of all possible interactions (28 out of 66) of GV proteins have been observed, indicating a dense network of protein–protein interactions (Fig. 3A–C; Dataset EV1). Notably, cells were assayed at two different time points in our experimental workflow, and a positive interaction observed in either time point will indicate an interaction between the two GV proteins. At both time points assayed, cells were post steady-state growth, but had not exhausted growth media (compare (Sezonov et al, 2007)), although the time points correlated with a significant difference in cell density. A grouped overview of the $OD_{600}$ of our measurements is given in Appendix Fig. S5A.

Next, we compared the interaction strength at the two different time points, 4 h ($t_4$) and 24 h ($t_{24}$) post-induction, respectively. We anticipated that $t_4$ and $t_{24}$ would reveal different states of protein–protein interactions, because $t_4$ corresponds to the initial assembly stage when GVs start to emerge in cells, while $t_{24}$ would correspond to the condition of fully assembled GVs (Lakshmanan et al, 2017). Among the interactions that showed a strong dependence on time (Fig. 3D–F), the major shell proteinGvpA2 and GvpG showed an increase in most of their interactions, whereas GvpJ and GvpU showed a decrease in most interactions. Notably, GvpF was not observed to interact directly with GvpA2 in our initial $t_4$ and $t_{24}$ datasets in contrast to previous reports (Völkner et al, 2020), and this was investigated further (see the section entitled "GvpA2–GvpF-GvpG form an interdependent interaction subnetwork").

## Targeted deletions of Gvps unravel protein interaction dependencies

Following this, we seek to determine whether an observed interaction is interdependent on a third protein (dependent interactions) or is not influenced by the absence of other proteins, which would suggest an independent (direct) interaction. This information can be critical for subsequent studies such as the construction of hierarchical interaction events and structural investigation of the binding sites. Since the roadmaps presented above were constructed in the presence of all 11 GV proteins, we anticipate that only a subset of the interactions were direct ones. Experimentally, we investigated only those pairs that had already shown an interaction in our previous experiments, except for the major shell protein GvpA2, of which we screened the interaction with all other GV proteins even if we had not seen an interaction. We also reasoned that keeping the 3-plasmid system will best retain a similar experimental condition for the cells such as the presence of all three antibiotics, and thus we created a decoy Plasmid III that contained no GV protein but only the plasmid backbone (ΔAll, Fig. 1).

Critically, we acknowledge that select protein–protein interactions in the GV operon may also be dependent on proteins natively present in *E. coli*. To limit the scope of this study and focus on the core essential GV expression system, we did not investigate this possibility in this study.

In the data (Fig. 4A–D), we observed both the cases where the removal of the background GV proteins diminished interactions as expected for the cases of dependent interaction pairs, e.g., GvpF and GvpN, and the cases where stronger interactions emerged after the removal of the other GV proteins, e.g., GvpA2 and GvpF, which were surprising and deserved more investigation (see the next section). Overall, the interactions involving GvpA2 and GvpJ saw a high number of increases. Since the major shell protein, GvpA2, has a highly hydrophobic N-terminal region that forms the inner surface of GVs (Huber et al, 2023), and GvpJ bears substantial homology in this hydrophobic region of GvpA2 (Fig. 4E), our data might imply that in the absence of other GV proteins to possibly chaperone or assembly GvpA2 and GvpJ, their interactions with any given binding partner would become stronger. In contrast, GvpS, which is the only other protein in the operon that has sequence homology to GvpA2, showed two strongly reduced interactions together with a couple of weakly increased interactions after the deletion of the background GV proteins. The contrasting

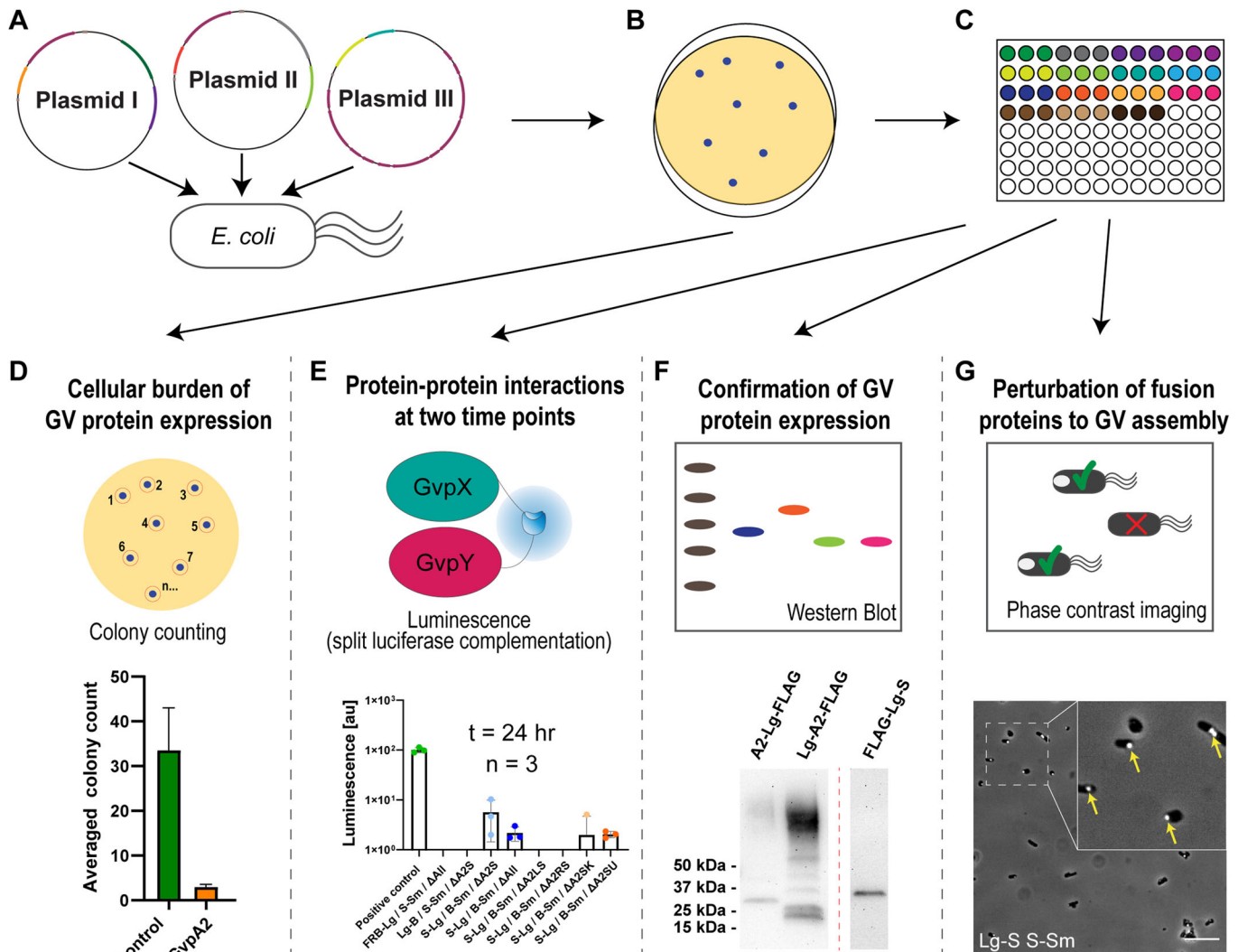

**Figure 2. Workflow to maximize information gained from protein–protein interaction screening.**

(A) Three plasmids from groups I, II, and III are selected and co-transformed into *E. coli* for each condition to be tested. (B) After plating transformants, colony formation is observed. Each plate represents a different protein–protein interaction condition. (C) Three colonies are selected for liquid culture from each plate and used for the following assays: (D) For most combinations of plasmids transformed (indicated in Table EV2), colonies are counted to confirm relative transformation efficiency (top; cartoon depicts a bacterial agar plate with colonies being counted). As an example, the average colony count for transformations including only the positive control, or GvpA2 fusion proteins is provided (bottom; full dataset in Fig. 6A). For colony counting, at least $n = 20$ transformations per condition were considered. Tails indicate SD. (E) Luminescence as a function of split-Nanoluciferase complementation was measured for each triplicate sample at $t_4$ and $t_{24}$ (top; cartoon depicts split-luciferase complementation). Luminescence values are documented as a percentage of the positive control and exemplary values are provided for plasmid combinations indicated on the x axis (bottom; full dataset in Dataset EV1). Tails indicate SD, dots indicate individual repeats (values below 0 due to normalization are not plotted). The bars indicate the mean. (F) Western blot was carried out for select samples (indicated in Dataset EV1) to detect the FLAG-tag of Large BiT-Gvp fusion proteins and confirm expression (top; a sketch of a western blot experiment). An example for western blot detection of A2-Lg-FLAG, FLAG-Lg-A2 and FLAG-Lg-S is provided (bottom; overview of western blotting results in Appendix Figs. S2 and S4). The data shown for Lg-A2/A2-Sm/ΔAll is identical to the data shown in Fig. 6D. (G) For select samples (indicated in Appendix Fig. S4), samples were analyzed by phase-contrast microscopy. A sketch depicts how the presence of a bright spot in phase-contrast microscopy signals GV-positive cells (top). An example of cells containing the plasmids Lg-S, S-Sm, and ΔS testing GV-positive under phase-contrast microscopy is provided and yellow arrows point to instances of GVs being detected (bottom; scale bar is 20 μM; complete dataset of GV-positive samples in Appendix Fig. S4). Experiments were carried out as described in the methods section. Source data are available online for this figure.

behavior of GvpS, GvpJ, and GvpA2 was further corroborated by the biochemical data that showed cellular burden only from GvpA2 and a tolerance of GvpS fusion for GV assembly (see the section entitled "A fraction of Gvp fusion proteins enable GV formation but affect cell viability"). Together, these data suggested different behavior of GvpJ and GvpS compared to GvpA2 despite the homologous sequence.

## GvpA2–GvpF–GvpG form an interdependent interaction subnetwork

Next, we proceeded to probe the subnetworks of interactions among the few intriguing clusters of proteins identified from the above roadmaps. Experimentally, we created a set of triple deletion Plasmid III that contained the GV operon excluding the two

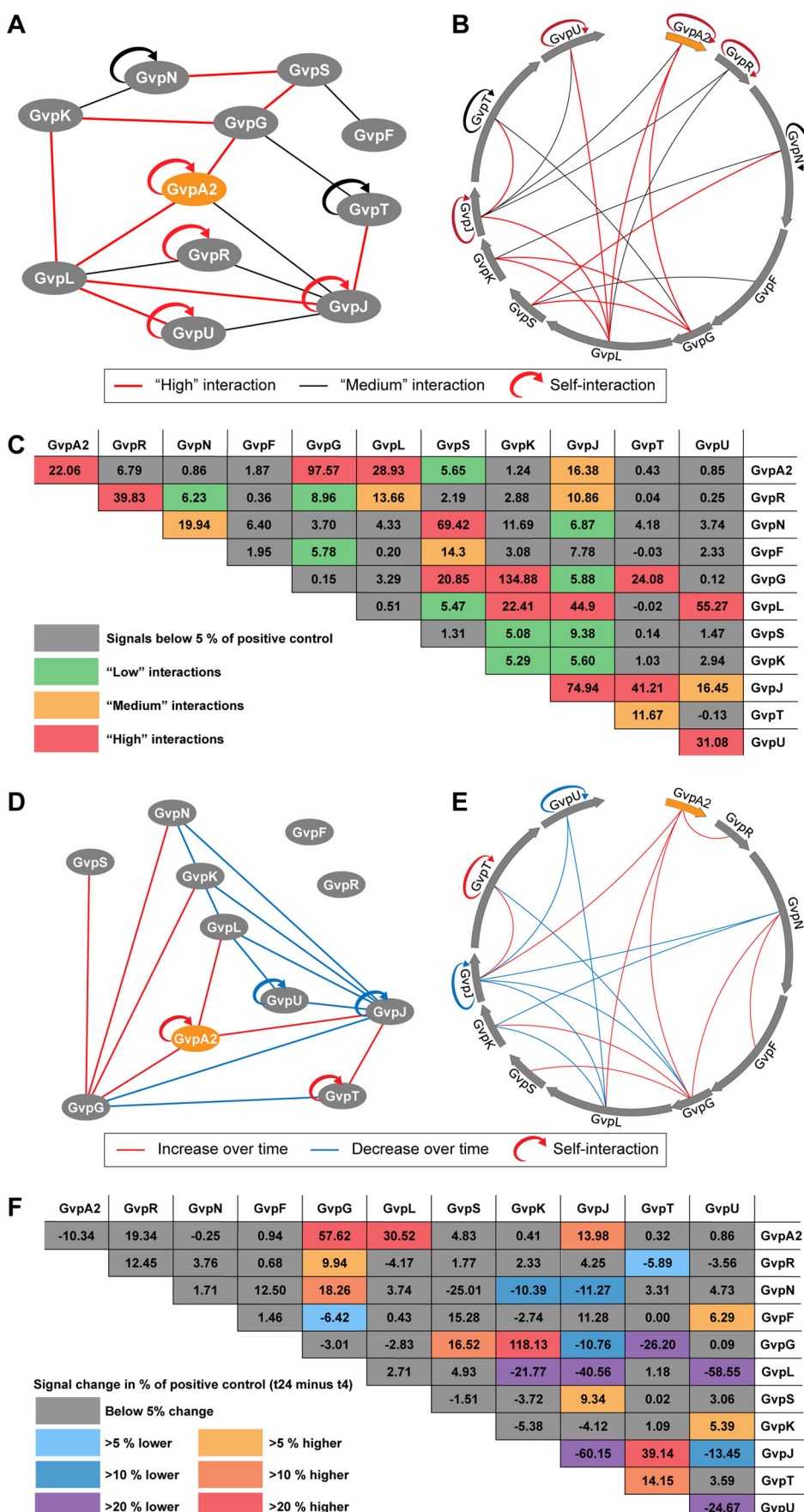

◀

**Figure 3. Systematic protein–protein interaction roadmaps of the GV operon.**

(A, B) Two styles of representations of the GV protein interaction network. The major shell protein GvpA2 is represented as an orange oval, and all other GV proteins as gray ovals. (B) GV proteins are displayed in a circle extracted from the DNA sequence map of the pNL29 GV operon with the actual order and length of the GV proteins preserved. The interactions between proteins are indicated by black lines if they are higher than 10% ("medium" interaction) of the positive control and red lines if higher than 20% ("high" interaction) of the positive control. Half-circular arrows represent the observed self-interactions, which indicate the oligomeric states of the proteins. (C) A summary table of the GV protein interactions. For each pair of GV proteins, the highest measured interaction in % of positive control is plotted. Each cell in the table is color-coded to differentiate the four categories of interaction strengths as labeled in the legend. Datapoints that did not prove statistically significant are provided, but with gray background. (D, E) Representations of the change of interaction network over time. A red line is used for interactions where the signal is increased by at least 10% of the positive control (in absolute terms) at $t_{24}$ over $t_4$, and a blue line if the signal is decreased by at least 10%. (F) A summary table of the interactions changed over time. Yellow to red colors are used to label an increase in interaction strength, and blue to violet colors for a decrease in interaction strength, as labeled in the legend. Datapoints that did not prove statistically significant are provided, but with gray background. Protein–protein interaction studies were carried out in $n = 3$ biological replicates. See also the full list of interaction data in Dataset EV1. Source data are available online for this figure.

proteins of interest and a third protein that is hypothesized to mediate their interaction. This strategy allows us to probe the interdependence of certain protein–protein interactions on individual GV proteins. Notably, in our initial interaction screening, we did not observe an interaction between GvpA2 and GvpF (Fig. 3A–C), contrary to a previous report that GvpF was the sole interaction partner of the major shell protein (Völkner et al, 2020). Surprisingly, a strong interaction of GvpA2 and GvpF re-emerged upon deletion of all other proteins in the GV operon (Fig. 4D). This hints at the possibility that one or a few of the GV proteins may interfere with the GvpA2–GvpF complex, contrary to our assumption that a third protein usually bridges the formation of a complex. To uncover which GV protein is playing this role, we first compared those proteins that interacted with both GvpA2 and GvpF and found out GvpG had the highest overall interaction (Fig. 3C). We then created a new plasmid, ΔBFG, of the GV operon with triple deletion of GvpA2, GvpF, and GvpG, and indeed, the deletion of GvpG was sufficient to resurrect the GvpA2–GvpF interaction (Fig. 5A), which strongly suggested that GvpF was outcompeted for binding of GvpA2 in the presence of GvpG. In parallel, another intriguing observation made in the last section was that, while GvpA2 saw primarily an increase in interactions when other GV proteins were removed, its interaction with GvpG was the only one that decreased (Fig. 4C). Thus, we hypothesize that GvpA2–GvpF interaction may be a prerequisite to the interaction of GvpA2–GvpG. We went on to test the interaction of GvpA2 and GvpG in the presence of all other GV proteins except for GvpF and observed that the deletion of GvpF alone was sufficient to attenuate the interaction to a level similar to the ΔAll condition. From these two results, we postulate that GvpF is the primary interaction partner of GvpA2, and the formation of the GvpA2–GvpF complex recruits GvpG, which in turn supersedes the binding of GvpF (Fig. 5B).

It is interesting to compare our observations with a previous study on GV proteins from haloarchaea, which did not include other GV proteins in the background (Völkner et al, 2020). For the BFG interaction subnetwork, this study observed the interaction of the shell protein GvpA with GvpF, but not with GvpG. The GvpA–GvpF interaction is in agreement with our GvpA2–GvpF interaction in the absence of background GV proteins, and the reported absence of the GvpA–GvpG interaction is also consistent with our finding that GvpA2 and GvpG do not interact in the absence of background GV proteins. However, the presence of the GV operon significantly altered the outcome of the interactions, leading to the discovery of the intriguing interdependent relation

among the three proteins. This highlights the importance of modulating background GV proteins in studying the interaction of GV gene clusters.

## The GvpA2–GvpL interaction is dependent on GvpS, K, and J

Finally, we investigated the observation that GvpA2 and GvpL do not interact at timepoint $t_4$ but interact strongly at $t_{24}$, as this is one of the pairs that showed a substantial time-dependent change of interactions (Figs. 3F and 5C). Also, the $t_{24}$ dataset revealed that deleting background GV proteins will substantially attenuate GvpA2–GvpL interaction (Fig. 5C). Thus, we hypothesized that additional GV proteins may mediate the GvpA2–GvpL interaction, and this interaction may also be dependent on the stage of GV assembly. Experimentally, we individually removed four candidate proteins and created the triple deletion plasmids of ΔBLS, ΔBLK, ΔBLJ, and ΔBLR. Except for GvpR, deleting any one of the other three proteins abolished the GvpA2–GvpL interaction (Fig. 5C), indicating that GvpS, GvpK, and GvpJ all mediate the interaction between GvpA2 and GvpL. This is further corroborated by the observation of strong interaction of GvpL with GvpS, GvpK, and GvpJ especially at the early time point ($t_4$) (Fig. 3C,F). Thus, we postulate that GvpA2 or GvpL first interacts with GvpS, GvpK, and GvpJ, and the resulting protein complex may be essential for the GvpA2–GvpL interaction at a later stage of GV assembly (Fig. 5D). Notably, among this protein cluster, GvpL was modeled to have a similar structure as GvpF (Winter et al, 2018), while GvpA2, GvpJ, and GvpS have sequence homology. Thus, based on the previous observation that GvpF interacts with GvpA2, it is reasonable to speculate that GvpL may interact with GvpA2, GvpJ, and GvpS in a similar fashion. However, this would leave the interesting question of why there are homologous yet essential proteins in the GV operon and what different roles they may play during GV assembly.

## A fraction of Gvp fusion proteins enable GV formation but affect cell viability

As a first addition to our dataset, we hypothesized that the number of *E. coli* transformants per protein set assayed may reflect the cellular burden of the GV proteins being expressed, since leaky expression can still occur without chemical induction during the growth of cells on agar plates. Among GV proteins, the major shell protein, GvpA2, has a highly hydrophobic surface, which is known to cause aggregation and proteotoxicity (Jung et al, 2021). Indeed,

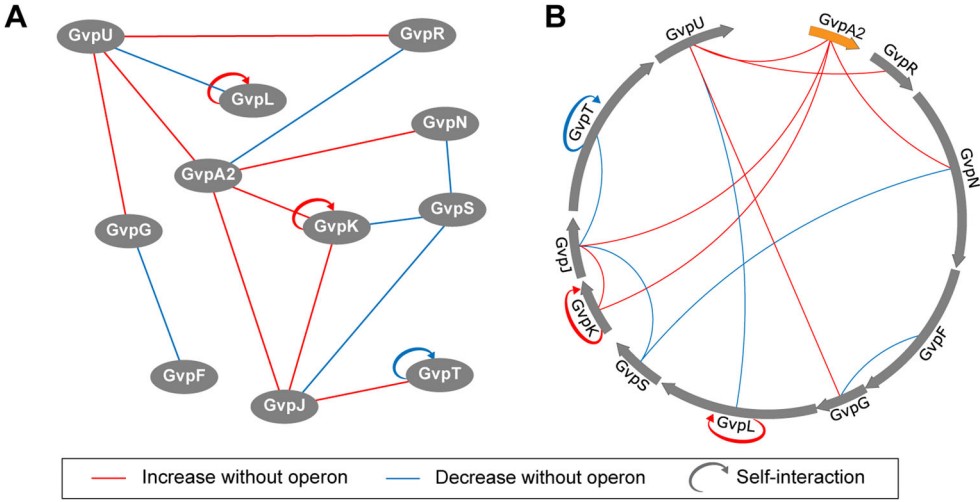

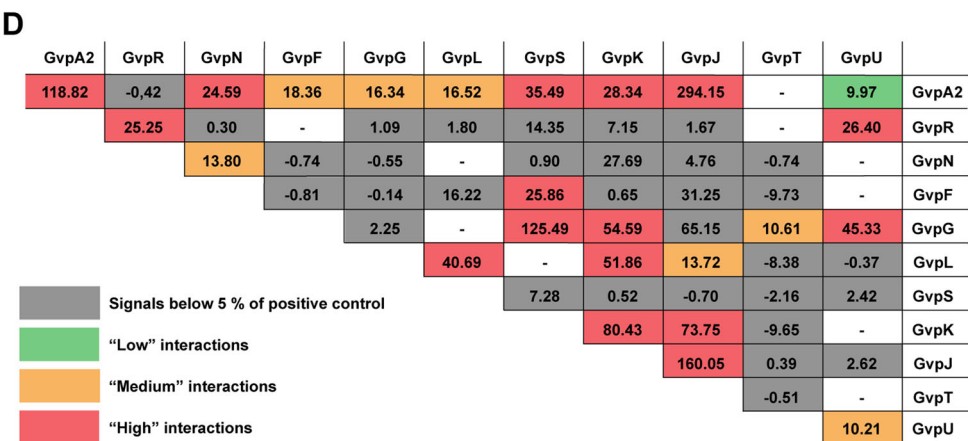

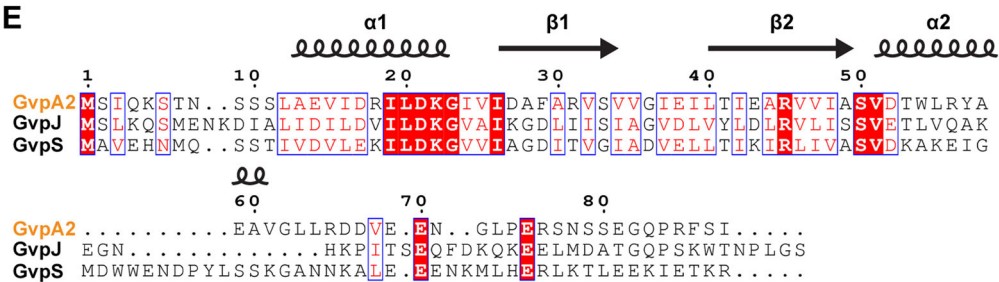

**Figure 4. Distinguishing dependent versus independent interactions of GV proteins.**

(A, B) Two styles of representation of the change of protein–protein interactions when the background GV proteins are removed. Blue lines indicate interactions that are at least tenfold lower in the absence of background GV proteins, and red lines indicate at least tenfold higher. (C) A summary table of the protein interaction fold changes in the absence of the background GV protein. The fold change is calculated as the signal in the presence of the background GV protein divided by that in the absence. Datapoints are omitted if interaction strength in neither condition is above 5%, since dividing datapoints of low interaction strength can generate spurious high fold changes. Yellow to red colors are used to label an increase in interaction strength, and blue to violet colors for a decrease in interaction strength as labeled in the legend. Datapoints that did not prove statistically significant are provided, but with gray background. (D) A summary table of the normalized interaction strength signal in the absence of the background GV protein. Each cell in the table is color-coded analogous to Fig. 3C to differentiate the four categories labeled in the legend. Datapoints that did not prove statistically significant are provided, but with gray background. Protein–protein interaction studies were carried out in $n = 3$ biological replicates. (E) Protein sequence alignment of the major shell protein GvpA2 and two assembly factors, GvpJ and GvpS. Source data are available online for this figure.

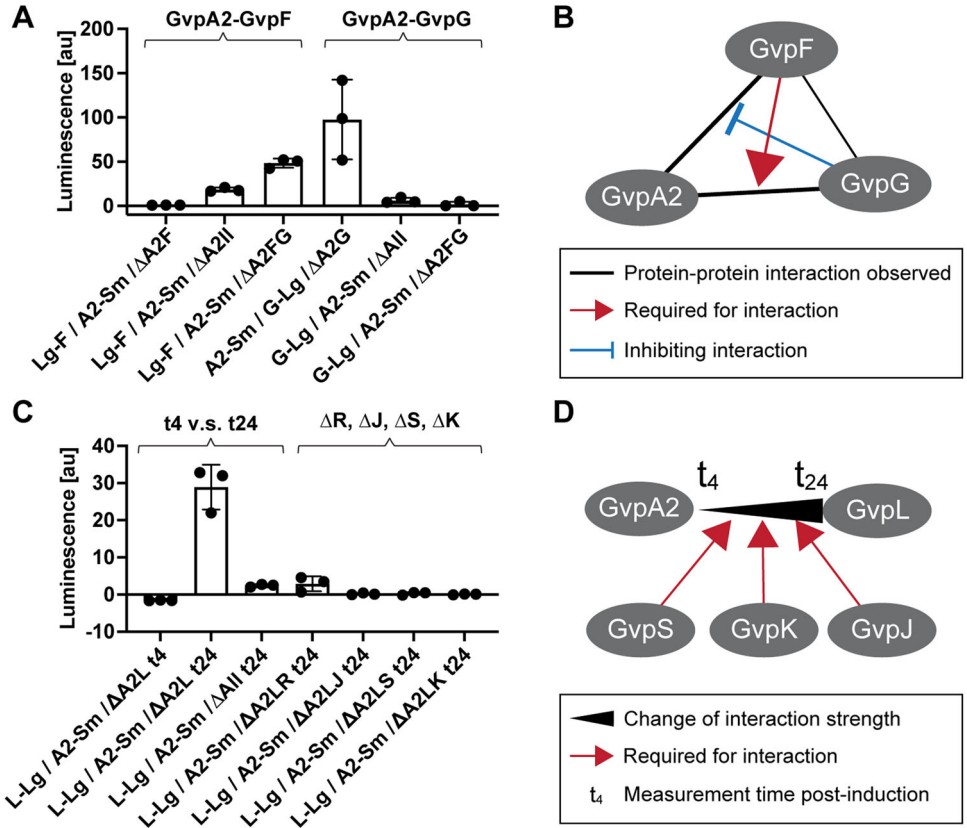

**Figure 5. Targeted deletion of GV proteins reveals interaction subnetworks in the GV operon.**

(A) Protein–protein interaction measurements for the subnetwork of GvpA2–GvpF–GvpG. The interactions between GvpA2 fused to Small BiT (A2-Sm) and GvpF fused to Large BiT (Lg-F) were measured in the presence of all other GV proteins (ΔBF), in the absence of other GV proteins (ΔAll), and in the presence of all other GV proteins except for GvpG (ΔA2FG). Similarly, the interactions between A2-Sm and GvpG fused to Large BiT (G-Lg) were measured in the presence of all other GV proteins (ΔA2G), in the absence of other GV proteins (ΔAll), and in the presence of all other GV proteins except for GvpF (ΔBFG). Tails indicate SD. The bars indicate the mean. (B) The postulated interaction map of GvpA2–GvpF–GvpG. The types of interactions are labeled in the legend below the map. (C) Protein-protein interaction measurements for the subnetwork of GvpA2–GvpL. The interactions between A2-Sm and GvpL fused to Large BiT (L-Lg) were observed in the presence of all other GV proteins (ΔBL) at 4 h and 24 h post-induction ($t_4$ and $t_{24}$), in the absence of all other GV proteins at 24 h post-induction (ΔAll t24), and in the presence of all other GV proteins except for single knockout of GvpR (ΔA2LR), GvpJ (ΔA2LJ), GvpS (ΔA2LS), and GvpK (ΔA2LK). The interactions are separated into two groups labeled on the top of the graph for the investigation of the change of interaction over time ($t_4$ vs $t_{24}$) and the investigation of the dependence of the interaction on individual GV protein (ΔR, ΔJ, ΔS, ΔK). Tails indicate SD. The bars indicate the mean. (D) The postulated interaction map of GvpA2–GvpL. Protein–protein interaction studies were carried out in $n = 3$ biological replicates. Source data are available online for this figure.

we observed that every transformation that included expression of a Gvp fusion protein led to a decrease in the total number of transformants compared to the control, with GvpA2 standing out as having the lowest total colony count (Fig. 6A; Table EV2). The other Gvps also showed a decrease of transformation efficiency,

indicating that the expression of gas vesicle proteins induces more cellular burden than the control proteins. It is surprising that overexpression of GvpJ and GvpS, which bear high sequence homology to GvpA2, do not have the same impact on transformation efficiency. Based on our colony counting dataset, it appears

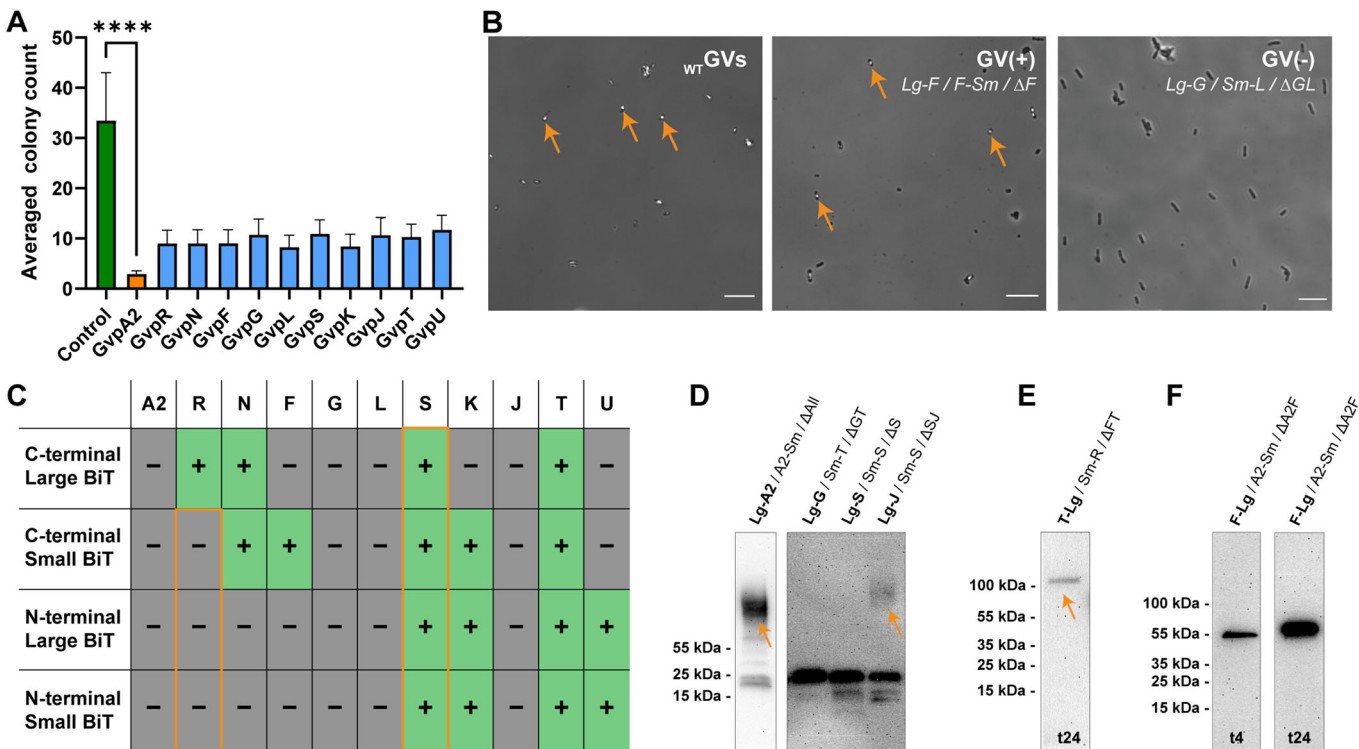

**Figure 6. Metabolic burden of Gvp expression, tolerance of fusion proteins in GV assembly and expression control of Gvps.**

(A) The colony counts after transformation indicate cellular burden caused by expressing each GV protein. The colony counts are binned into three groups, and at least three biological replicates and both N- and C-terminal fusion forms are included for each protein. See the full list of $n = 533$ counted plates in Table EV2. One-way ANOVA testing was performed followed by Dunnett's multiple comparisons test to confirm significance vs. the control (testing all samples vs a single control group). $P$ values for colony count vs control are from left to right: GvpB 0.0001; GvpR 0.0004; GvpN 0.0004; GvpF 0.0002; GvpG 0.0008; GvpL 0.0001; GvpS 0.0009; GvpK 0.0002; GvpJ 0.0014; GvpT 0.0004; GvpU 0.0011. Tails indicate SD. The bars indicate the mean. (B) Example phase-contrast images of E. coli. Orange arrows indicate the representative *E. coli* cells that contain successfully assembled GVs, which are usually revealed as intracellular bright spots. Scale bars represent 20 µm. (C) A summary table of whether successfully assembled GVs were observed in phase-contrast images for all four types of fusion constructs and all the GV proteins, providing clues for whether a specific fusion will perturb the GV assembly process. See also Table EV2 for the full list of all the conditions tested for phase-contrast imaging and Appendix Fig. S3 for representative phase-contrast images of all the conditions containing successfully assembled GVs. (D) Example western blots of the high molecular weight smeared bands of GvpA2 and GvpJ (labeled by orange arrows). The data shown for LG-A2/A2-Sm/ΔAll is identical to the data shown in Fig. 2F. (E) Western blot of dimer GvpT-Lg fusion proteins (labeled by an orange arrow). (F) Western blots of the GvpF-Lg fusion protein at $t_4$ and $t_{24}$, representing an example of the increasing protein expression level over time. See also Appendix Fig. S2 for additional western blot images of GV protein–Large BiT fusion used in this study. Source data are available online for this figure.

that GvpJ and GvpS do not form proteotoxic protein species to the same degree as GvpA2 does. This finding is supported by the very notable presence of SDS-insoluble high molecular weight species for GvpA2 (Appendix Fig. S2).

Following this, we assessed whether GV assembly is compatible with the individual Gvp fusion proteins. Making genetic fusions is a common tool to investigate the function, localization, and structure of proteins; however, this also risks perturbing the native function of the protein by the fused partners. Since we have already created all the constructs with both N- and C-terminal fusion of either the Large BiT, a large globular protein, or the Small BiT, a short 11-amino-acid peptide, we sampled our protein–protein interaction assay to check for GV assembly. Experimentally, the successful assembly of GVs can be determined by phase-contrast imaging of bacteria, since the gas compartments of GVs scatter light and produce a distinguishable white spot on these images (Farhadi et al, 2020) (Fig. 6B). We focused only on conditions where all the GV proteins were provided in the cell and for each configuration of GV fusion proteins, we sampled at least three independent cultures at

$t_{24}$. A successful assembly should be observed if two conditions were met: (i) the presence of the fusion partner did not interfere with the function of proteins and (ii) the expression level of the fusion proteins does not perturb the stoichiometry of the GV proteins to be beyond the range tolerated by the assembly process. Notably, (i) can also be satisfied if the target GV protein is not essential for GV formation. Since our experiments always have two of the GV proteins simultaneously fused to Large BiT and Small BiT, the observation of assembled GVs will indicate that both fusion proteins were permissive to GV formation, whereas no GVs observed indicated that at least one of the fusion proteins interfered with GV formation. Screening through 149 conditions led to the discovery that 17 conditions permitted the formation of GVs, and grouping these data allowed us to identify which fusion proteins were suitable for GV formation by elimination (Fig. 6C; Table EV3). First, we observed that all fusion proteins of GvpS and GvpT permitted GV formation. While GvpT was known to be non-essential to the formation of GVs (Farhadi et al, 2019), it was surprising to see that an essential protein, GvpS, allowed for GV

formation to a level close to the wildtype in all the fusion configurations. Moreover, GvpS bears high homology to GvpA2 on the N-terminal hydrophobic region (Fig. 4E), and while none of the GvpA2 fusions gave intact GVs, all GvpS ones permitted the assembly. Another surprising result was that, while GvpR was previously determined to be non-essential (Farhadi et al, 2019), we observed that 3 out of the 4 fusion configurations of GvpR would perturb GV formation. Thus, the creation of less-functional GvpR might cause more disturbance to the assembly process of GVs than simply eliminating GvpR. We anticipate these findings would pave the way to uncover the functional role of GvpR in the assembly process of GVs. For all other proteins, the site of fusion and size of the split-luciferase fragment influenced GV formation as expected, and these results will guide the future design of fusion proteins to study the function and cellular location of these proteins.

### All Gvp fusion proteins are expressed but vary in stability

Lastly, to monitor the expression levels of our fusion protein constructs and to exclude that an absence of luciferase signal is due to poor protein expression, performed western blot immunodetection of FLAG-tags on our fusion proteins. For each N- and C-terminal Large BiT fusion protein, we analyzed three independent samples (Dataset EV1 indicates which interaction experiments were sampled). We observed that all fusion proteins were expressed and appeared at the expected molecular weight with the exception of the major GV shell protein GvpA2 and its homolog, GvpJ. They were observed to also produce smeared bands at high molecular weight, suggesting SDS-insoluble oligomers (Fig. 6D), and this agrees with previous studies on SDS-PAGE of GV major shell protein (Walsby and Hayes, 1988). We also observed a dimer form of GvpT (Fig. 6E), which agrees with the observation of strong self-interaction of GvpT at $t_{24}$. Lastly, we observed a general trend for higher expression levels at $t_{24}$ than $t_4$, in line with the higher raw luminescence intensity at $t_{24}$ (Fig. 6F). Notably, higher expression levels at $t_{24}$ also hold true for the positive control samples, supporting the choice of using the percentile rather than raw luminescence intensity as the quantification method. Moreover, we observed that some of our fusion proteins had undergone truncation or partial proteolysis, particularly when sampled at the $t_{24}$ timepoint (for example, Lg-L, Lg-K, or Lg-T in Appendix Fig. S2). The accumulation and possible digestion or denaturation of fusion proteins over time may confound a part of our protein interaction data by introducing aberrant, non-physiological interactions. Consequently, our dataset must be seen as semiquantitative, confirming our decision to bin datapoints as "high", "medium" or "low", with no absolute quantification of individual levels of protein–protein interaction. Uncropped versions of the western blots showing in Appendix Fig. S2 can be seen in Appendix Fig. S4.

## Discussion

Despite their emerging biomedical value, and recent progress in understanding their structure (Dutka et al, 2023; Huber et al, 2023), the assembly process of GVs remains largely unclear. In this work, we established a systematic protein–protein interaction roadmap of gas vesicle proteins that revealed a dense interplay of proteins and

included the interdependence of the gas vesicle protein interactions with each other.

Such a protein–protein interaction network is often one of the first steps to understanding the assembly mechanism of a protein organelle. For example, delineating the protein–protein interactions of β-carboxysome biogenesis revealed the core-first assembly pathway (Cameron et al, 2013), which paved the way to many subsequent studies on the structure and function of individual carboxysome proteins, the redesign of the enzymatic core and shell protein scaffolds, and engineering of carboxysome-like organelles for metabolic engineering (Kerfeld et al, 2018; Li et al, 2020; Wang et al, 2019). Similarly, we anticipate that this GV protein interaction roadmap will lay the foundation for future studies and engineering of GVs. Specifically, understanding the protein interactions during GV assembly will guide the optimization of the heterologous expression in therapeutically relevant cells by informing the interdependence of protein players, of which the stoichiometry may be aligned, and the binding interface may need to be optimized in a particular cellular context. The poor assembly efficiency during heterologous expression is currently the main obstacle hindering the biotechnological applications of GVs, exemplified by the finding that several assembly factor proteins need to be supplemented in a "booster" plasmid during the mammalian expression of GVs (Farhadi et al, 2019). In addition, this work provided systematic information on the tolerance of GV proteins for terminal fusions, which may be the sites for adding designed proteins to GVs. Finally, constructing the protein interaction roadmap in this work will be an essential step toward in vitro reconstitution of GVs, which will overcome the current limitation that GVs have to be manufactured inside a cell. While the current understanding of the biology of GVs is still in its infancy, we anticipate a rapid growth of studies on this topic to match the recent surge of biotechnological applications of GVs.

Building on current knowledge and this study, we would like to postulate an assembly process of GVs encoded in the pNL29 operon (Fig. 7). Stage I would be the initial "seeding" of GVs, which is currently the least understood step in the entire assembly process due to the difficulty of experimentally observing these seeding GV protein complexes. Here we propose that GvpS, GvpK, and GvpJ may be involved in this seeding process of GVs. This is supported by the observation that these three proteins are required for the interaction between GvpA2 and GvpL (Fig. 5D), hinting that they act upstream of the major shell protein GvpA2 in the generation of new GVs. Corroborating this observation, the interaction of *H. salinarum* Gvps M (homolog to GvpS), K and J with GvpL, which has also been reported and proposed as a type of scaffold protein (Völkner et al, 2020).

The earliest observable stage of GV growth is the "bicone" (Pfeifer, 2022; Walsby and Hayes, 1989), which we labeled as Stage II, and GvpS, GvpK, and GvpJ likely continue to be involved. Among them, GvpK was notably missing from having a physical attachment to GV particles in a previous mass spectrometry study (Chu et al, 2011), suggesting that GvpK is fully soluble in the cytosol, and this corroborates our observation that most GvpK fusion proteins do not interfere with GV assembly (Fig. 6C). Notably, GvpK is one of the few proteins consistently grouped as essential in all 4 major genotypes of GVs studied to date (Farhadi et al, 2019; Hurt et al, 2023; Offner et al, 2000; Tashiro et al, 2016), and yet little information is present to suggest its functional role,

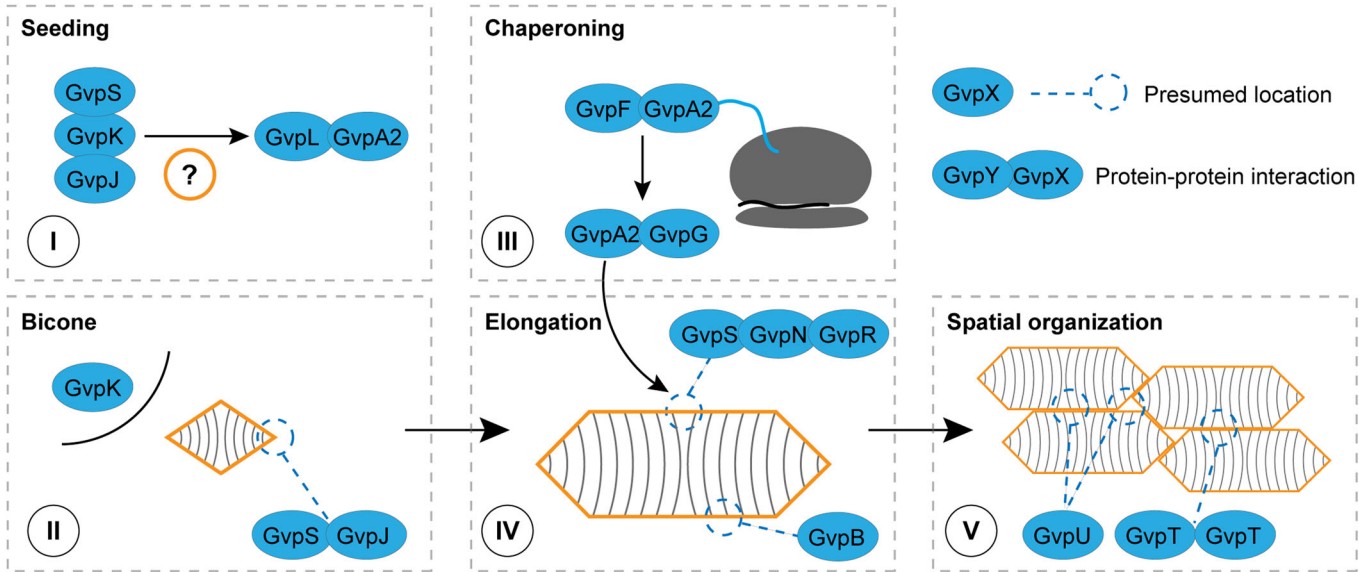

**Figure 7. Proposed involvement of GV proteins in each stage of the assembly.**

The five stages are drawn into five panels with dashed-line borders. The stage number is indicated by a Roman numeral in a black circle, and the stage name is labeled in the top left corner of each panel. Proteins are colored in blue, and the protein shell of GVs is indicated with orange color. A schematic of a ribosome is colored grey in Stage III.

and our observation that the presence of GvpK is essential for the GvpA2–GvpL interaction (Fig. 5D), as well as other notable interactions with GvpN and GvpG, may lead to possible directions to investigate its function. Different from GvpK, GvpS, and GvpJ consist of sequences homologous to the highly hydrophobic, inside-facing α1-β1-β2 segment of GvpA2, and thus GvpS and GvpJ may not be fully soluble in the cytosol. While GvpA2 can fully occupy the middle segment of GVs according to the cryo-EM structures (Dutka et al, 2023; Huber et al, 2023), GvpS and GvpJ may occupy the tips of newly formed GVs, and this hypothesis was strongly supported by a recent study on purified bicone GVs, which showed the presence of GvpJ and GvpS in the particles even after washing with 6 M urea (Ling et al, 2023).

The "chaperoning" of GvpA2, Stage III, is likely to be carried out by GvpF as the primary binding partner of GvpA2 (Figs. 4D and 5A), an interaction that is superseded by the secondary binding of GvpG (Figs. 3C and 5B). We hypothesize that GvpF and possibly GvpG may act as chaperones of GvpA2 and prevent it from becoming thermodynamically trapped and forming proteotoxic protein species (Jung et al, 2021) out of a number of considerations. For *H. salinarum* GvpF, it was observed that even the truncated α1 helix of GvpA suffices as a binding partner, indicating early, possibly cotranslational, binding of GvpF as was also suggested by Pfeifer, 2022 (Pfeifer, 2022). A structural study of GvpF from *M. aeruginosa* further indicated that GvpF binds the interior (to the GV), hydrophobic portion of the shell protein (Xu et al, 2014). The presence of GvpG attenuates the binding of GvpF to GvpA2 (Fig. 5A,B), indicating that either they compete for the same binding surface or the fully translated GvpA2 adopts a conformation with higher affinity to GvpG. This further suggests that the interaction between GvpF and GvpA2 is transient, which in combination with binding of the hydrophobic portion of the major shell protein is our basis for suggesting a chaperone-like interaction

(Hartl et al, 2011). It is worth noting that GvpF and GvpG also interact with GvpS and GvpJ and may participate in Stage (I) and (II) as well.

For the elongation of GVs (Stage IV), it is likely that the insertion of the shell protein GvpA2 occurs at the middle polarity reversal point observed in the cryo-EM structures (Dutka et al, 2023; Huber et al, 2023), and yet how the assembly factor proteins facilitate the insertion process remains unclear. GvpN likely participates, since it is established that ΔGvpN mutants allow the formation of bicone GVs but these GVs do not elongate in the absence of GvpN (Jost and Pfeifer, 2022; Ling et al, 2023; Monson et al, 2016; Offner et al, 2000). GvpN is an AAA+ ATPase, and we observed its self-interaction (Fig. 3C), which is in agreement with AAA+ protein-specific homooligomers (Mogk et al, 2008). AAA+ ATPases are known to promote degradation or refolding of their protein substrates (Mogk et al, 2008), and we hypothesize that GvpN uses energy to prepare the correct folding of GvpA2 for insertion into the growing cylindrical portion of GVs, or alternatively, uses energy to proofread the folding of GvpA2 along GV particles (Matsumoto, 2023). The activity of the AAA+ ATPase domain in GvpN has been confirmed for a homolog in Anabaena sp. PCC 7120 (Cai et al, 2020). Since GvpA2 is the main component of GVs, our observation that GvpN interacts with GvpA2 and not GvpJ or GvpS in the absence of all other assembly factors (Fig. 4C) further support the role of GvpN in the elongation process of GVs. Interestingly, in the presence of all GV proteins, GvpS became the strongest interaction partner of GvpN (Fig. 3C), and thus GvpS might also participate in the growth of GVs at the polarity reversal point. Lastly, this interaction of GvpN and GvpS is partially dependent on the presence of GvpR (Dataset EV1), indicating that GvpR may assist the activity of GvpN. The relation between GvpR and GvpN is corroborated by previous reports that GvpO, a haloarchaeal homolog of GvpR, will influence the expression of

GvpA, GvpC, and GvpN (Chu et al, 2011; Offner et al, 1996). However, whether GvpR/O is essential for the assembly process remains controversial and may depend on cellular context: GvpO was grouped as an essential protein in haloarchaea (Offner et al, 2000), but for pNL29-encoded GVs, ΔGvpR mutant can produce GVs (Farhadi et al, 2019); and in this work, we additionally observed that most fusion proteins of GvpR would produce GV-negative cells (Fig. 6C) indicating an unproductive gain-of-function.

Finally, Stage V is the spatial organization of GVs. While this stage is not essential for the formation of GV particles, spatial organization would be important to minimize GVs' occupancy of cytosolic space and was recently shown to modulate cellular fitness (Li et al, 2024). GvpU and, to a lesser extent, GvpT mediate the clustering of GVs (Li et al, 2024), and both of them are non-essential proteins in the pNL29 operon (Farhadi et al, 2019). Intriguingly, we still observe interactions between GvpU, L, and J as well as GvpT, G, and J, which link them to the initiation and chaperoning stages of GV formation. One plausible explanation is that the clustering of GVs occurs concurrently with the assembly of GVs, and some of the assembly factor proteins may have even evolved affinity to GvpU and GvpT, which will help their spatial recruitment to the sites of GV assembly in the cytosol. In addition, we observed other notable interactions, including a strong interaction between GvpG and GvpK (Fig. 3C) which is unaffected by deleting the rest of the GV operon (Fig. 4C). Following the abovementioned binding partners of GvpT and GvpU, we furthermore observed that in the absence of all other GV proteins, GvpU shows an increase in binding to GvpR and GvpG (Fig. 4C). Together, these findings indicate that the stages of GV initiation, growth, chaperoning, and spatial organization are not discrete, but may occur in parallel and be organized by crosstalk between proteins. The resolution of the assembly mechanism of GVs will not only progress our understanding of how cells assemble complex protein structures but also mark a highly significant stride for this class of microbial organelles with remarkable physical properties.

This work must be seen in the context of prior studies on the interactions and putative functions of Gvps and we provide additional, granular observations on this in the Appendix (Appendix Discussion section).

We conclude by pointing out limitations to our study: It is likely that the use of fusion proteins and non-physiological expression systems (Appendix Fig. S1) impacts predictions for the complete interaction network of the GV operon. False-positive signals may arise from unproductive interactions between overexpressed GV proteins. Notably, GvpB and GvpJ form SDS-insoluble aggregates (Fig. 6D), and their self-interactions increase in the absence of the GV operon (Fig. 4C). This behavior can also be observed for *H. salinarum* GvpB and J (Pfeifer, 2022) and it is possible that interactions between the shell proteins in absence of the GV operon are non-productive and proteotoxic (Jung et al, 2021). Conversely, some GV proteins show signs of protein degradation in our experiments (Appendix Fig. S2, particularly Lg-T, Lg-L, and Lg-K), which may lead to dysfunctional proteins, synthetic false-positive interactions, or attenuated interaction measurements. Unlike a previous analysis (Völkner et al, 2020), our investigation was carried out screening only in vivo protein–protein interactions, prioritizing completeness of screening over depth of analysis. Notably, this approach can not address the possibility that proteins

of the *E. coli* host cell play a role in the assembly of *B. megaterium* derived GVs (a similar effect has been described for yeast in (Jung et al, 2021)) and in fact, we have not investigated the entirety of possible interdependencies. It follows that our study leaves room for future investigations focusing on more detailed, mechanistical analysis of subsets of protein–protein interactions described here.

In summary, our dataset has to be seen in a semi-quantitative manner and provides a basis, but not an endpoint, for investigating the assembly of bacterial gas vesicles.

## Methods

### Plasmid construction

DNA plasmids used in this study were generated by Gibson assembly and site-directed mutagenesis utilizing Q5 Hot Start High-Fidelity enzymes, HiFi DNA Assembly Master Mix, and KLD Enzyme Mix (New England Biolabs (NEB), Ipswich, MA). Plasmids were generated based on the pET-26b(+) backbone to contain the large or small fragments of the NanoBiT luciferase (Large BiT or Small BiT) used for the split-luciferase complementation assay (Promega, Madison, WI, USA) (Dixon et al, 2016). Large BiT or Small BiT were fused at the N- or C-terminus with GV proteins, FRB, or FKBP12, the latter two of which were used as a rapamycin-inducible positive control of dimerization (Rivera et al, 1996). An 18-amino acid linker VSQGSSGGGGSGGGGSSG was used between the two fusion partners for all the constructs (Kuhlman et al, 1997). For each construct carrying the Small BiT, the origin of replication was replaced with the p15A origin (Plasmid group II). For those constructs carrying the Large BiT, the origin of replication was replaced with the CloDF13 origin, and the kanamycin resistance gene was changed to a spectinomycin resistance gene (Plasmid group I). For easy detection by western blot, all Large BiT fusion proteins contained a FLAG-tag at the N- or C-terminus of the fusion protein distal to the Gvp.

In parallel, deletion mutants of each GV protein in the pNL29 operon and combinations of two or three thereof were generated as indicated. The pNL29 operon was first cloned from the pST39-pNL29 plasmid (Addgene ID 91696), and the ORF of GvpK overlapping the ORF of GvpS was resolved by moving the start codon of GvpK to the downstream of GvpS, generating a new plasmid named as Mega' (Appendix Fig. S1A). Subsequently, single, double, triple, and full knock-out (an empty backbone, ΔAll) variants of the GV operon were generated from the pST39-pNL29 ("Mega") or the Mega' plasmid (Plasmid group III). All the plasmids are listed in Table EV1, and the DNA sequences of all the parts are listed in Dataset EV2. Primer design as well as graphical presentation of plasmid constructs was carried out using SnapGene (GSL Biotech LLC, San Diego, CA, USA).

### Transformation

For molecular cloning, bacterial transformations were carried out using NEB Turbo competent cells, and the transformed cells were isolated on agar plates with the respective antibiotics (50 μg/mL Kanamycin or 75 μg/mL Spectinomycin or 100 μg/mL Carbenicillin). For protein–protein interaction experiments, home-made competent BL21(DE3) cells were generated by Mix & Go *E. coli*

Transformation Kit (Zymo Research). For each transformation, 20 μL of competent cells were mixed with ~50 ng of each of the 3 plasmids as indicated. Cells were incubated for 30 min on ice before heat shock for 30 s at 42 °C in a water bath. Cells were left to recover on ice for 5 min before the addition of 200 μL SOC media in a 1.5 mL tube, followed by gentle shaking for 4 h to the rescue. The transformation mixture was plated on bacterial agar plates supplemented with 0.5-fold of the previously indicated antibiotics and 1% (w/v) glucose to grow overnight at 37 °C. Plates with successful transformants were sealed and transferred to 4 °C until further use. For transformants that failed to recover on agar plates, the transformation procedure was repeated, and instead of plating, the resulting mixtures were transferred to 5 mL LB liquid media with the same additives to grow overnight at 30 °C.

## Split-luciferase complementation assay

Cell cultures, luminescence measurements, and optical density at 600 nm ($OD_{600}$) measurements were all carried out in 96-well microplates to maximize the throughput. First, *E. coli* BL21(DE3) cultures transformed with the 3 indicated plasmids were inoculated from agar plates or liquid media to 1 mL of liquid media containing 0.5-fold of the three antibiotics and 1% (w/v) glucose. Three separate cultures were raised for each combination to ensure 3 biological replicates for the subsequent experiments. The 1 mL cultures were grown overnight at 30 °C in a sealed 96-deep-well storage plate (SSI Bio Inc.), shaking at 600 rpm. For each experiment, a triplicate of Luria Broth (LB) media without bacterial cells was included as negative control and used subsequently for luminescence and $OD_{600}$ measurement.

On the following day, 10 μL of these overnight pre-culture cells were transferred to 190 μL of LB media containing antibiotics and 0.2% glucose in a clear 96-well microplate (#3370, Corning, Corning, NY, USA) and left to grow at 30 °C and 600 rpm until all cultures reached $OD_{600}$ above 0.3 measured in a Biotek Synergy H4 Multimode (Agilent, Santa Clara, CA, USA) microplate reader. Isopropyl b-D-1-thiogalactopyranoside (IPTG) was added to a final concentration of 20 μM in all wells, and this was counted as time zero ($t_0$). At 3.5 h after induction, rapamycin (Biovision, Milpitas, CA, USA) in DMSO was added to a final concentration of 25 μM in wells containing cells with the FRB-Lg / FKBP12-Sm / ΔAll constructs as positive control. At 4 h and 24 h after induction, respectively ($t_4$ and $t_{24}$), 10 μL of assay cells were mixed with 10 μL of NanoGlo Live Cell Substrate (Promega, Madison, Wisconsin, USA) and 30 μL of LB to make up a total of 50 μL reaction mixture in an opaque, 96-well half-area microplate (#3694, Corning, Corning, NY, USA) for the luminescence measurement. Shortly before the luminescence reading, $OD_{600}$ of all samples was measured in the clear 96-well plate, and samples for SDS-PAGE and phase-contrast microscopy were collected as indicated.

## Data processing and evaluation

Data from the split-luciferase complementation assay are reported as follows. $OD_{600}$ and luminescence of the blank buffer control were treated as the baseline, and the values are subtracted from all other measurements. Next, for each well, the luminescence measurement was divided by the $OD_{600}$ measurement to normalize the signal for variations in cell density. For each microplate, a

positive control was included that contained cells expressing FRB-Lg/FKBP12-Sm/ΔAll plasmids and was supplemented with rapamycin 30 min before measurement. The signal from positive control was set to be 100%. Each microplate also contained two negative controls. The two negative controls consisted of cells expressing FRB-Lg with an arbitrary GV protein fused with Small BiT or FKBP12-Sm with an arbitrary GV protein fused with Large BiT. These negative controls are made assuming that GV proteins do not interact with FKBP12 or FRB protein. All measurements lower than the negative control were considered to represent non-interaction, and all measurements of the samples were reported as percentages of the positive control. We noted that some interactions measured were stronger than the positive control, giving rise to values >100%.

Protein alignment for Fig. 4E was performed using *Clustal Omega* (Madeira et al, 2022; Sievers and Higgins, 2014), and the figure was generated using *ESPript* (Gouet et al, 2003). The secondary structure of GvpA2 indicated above the protein sequence was extracted from the cryo-EM structure (PDB: 7R1C) (Huber et al, 2023).

## Phase-contrast imaging

For phase-contrast microscopy, 5 μL of the bacterial cultures at $t_{24}$ were transferred to object slides and mixed with a drop of Fluoroshield mounting medium (Sigma-Aldrich). Coverslips were added and samples were left to dry for at least 30 min before analysis on an Eclipse TI2 inverted microscope (Nikon, Melville, NY, USA) to identify cells containing GVs. For imaging, the 40× phase-contrast objective and 1.5× tube lens are combined to result in a 60-fold total magnification. The Ph2 condenser annulus was used, and the exposure was held at 700 milliseconds for all the images. Phase-contrast images were prepared for publication with *Fiji* (Schindelin et al, 2012).

## Western blot

In total, 30 μL of the indicated bacterial cultures were mixed with 30 μL of 2× Laemmli buffer and incubated at 98 °C for 5 min. Samples (using 25 μL each) including 5 μL PageRuler Plus (Thermo Fisher Scientific, Waltham, MA, USA) or Precision Plus Dual Xtra (Bio-Rad, Hercules, CA, USA) prestained protein ladder were applied to SDS-PAGE using 4–20% Mini-PROTEAN TGX precast protein gels (Bio-Rad, Hercules, CA, USA).

Western blots were carried out in a Trans-Blot Turbo system using Trans-Blot Turbo Mini 0.2 μm Transfer-Packs (Bio-Rad, Hercules, CA, USA) using the Standard setting on the instrument. Total protein staining was performed using Revert 700 Total Protein Stain (LI-COR, Lincoln, NE, USA) following the manufacturer's instructions and detected using a FluorChem M imager (Protein Simple, San Jose, CA, USA) by the Multifluor Red setting. Immunodetection was carried out by blocking PVDF membranes in 5% (w/v) fat-free dried milk powder followed by incubation overnight in rat anti-FLAG M2 antibodies diluted 1:500 (Agilent Technologies, Santa Clara, CA, USA). After a brief rinse in Tris-buffered saline with Tween™ 20 detergent (TBST), goat anti-rat IgG (H + L) HRP-conjugated secondary antibodies (Invitrogen, Waltham, MA, USA) were used in 1:5000 dilution to incubate membranes for 1 h. Membranes were washed 3× in TBST, followed

by TBS, before detection using Pierce ECL Western Blotting Substrate (Thermo Fisher Scientific, Waltham, MA, USA) with the FluorChem M imager using the "Chemi + Markers" setting and exposing for no more than 10 min. Western blot images were contrast optimized with *Fiji* (Schindelin et al, 2012).

## Statistical analysis

Statistical testing of the protein–protein interaction data presented in Figs. 3–5 and Dataset EV1 was carried out by performing Welch's $t$ test (assuming uneven variance between different measurements) of each measurement vs the corresponding negative control. Interactions that did not prove statistically significant were not counted as interactions, but the absolute value is provided. $P$ values of reported measurements are indicated in Dataset EV1. For each measurement, $n = 3$ biological replicates have been used. For datapoints derived from the measurements ($t_4$ vs $t_{24}$ or interaction changes in the absence of the GV operon), changes were not counted if one of the base interactions was not statistically significant, to avoid inaccurate assessment.

For the $OD_{600}$ range reported for measurements taken at $t_4$ and $t_{24}$ in Appendix Fig. S5, a two-tailed unpaired $t$ test was performed (**** indicating $P$ value 0.0001).

For Fig. 6A, colony counting, at least $n = 20$ transformations per condition were considered and one-way ANOVA testing was performed followed by Dunnett's multiple comparisons test to confirm significance vs. the control (testing all samples vs a single control group).

Statistical testing was carried out with Excel (Microsoft, Redmond, WA) and Prism (Graphpad Software, Boston, MA).

## Data availability

The set of plasmids generated for this study has been made available as Addgene deposit number 83964.

The source data of this paper are collected in the following database record: biostudies:S-SCDT-10_1038-S44318-024-00178-2.

## Peer review information

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

## Acknowledgements

The authors would like to thank Thyer lab for providing template plasmids encoding the Spectinomycin resistance gene and the p15A and CloDF13 origins of replication. The authors thank the Shared Equipment Authority (SEA) at Rice University for the access to core facilities and instruments. This work was supported by the Cancer Prevention and Research Institute of Texas (RR190081), the National Institutes of Health (R35GM155015, R00EB024600 and R21EB033607), the Welch Foundation (C-2069), G. Harold and Leila Y. Mathers Foundation (MF-2012-01314), and John S Dunn Foundation. MI acknowledges support from the German Research Foundation (DFG 511048568) Postdoctoral Fellowship.

## Author contributions

**Manuel Iburg**: Conceptualization; Resources; Data curation; Software; Formal analysis; Validation; Investigation; Visualization; Methodology; Writing—original draft; Writing—review and editing. **Andrew P Anderson**: Conceptualization; Resources; Data curation; Software; Formal analysis; Validation; Investigation; Visualization; Methodology; Writing—original draft; Writing—review and editing. **Vivian T Wong**: Resources; Data curation; Software; Formal analysis; Investigation; Methodology. **Erica D Anton**: Resources; Data curation; Software; Formal analysis; Investigation; Methodology. **Art He**: Resources; Data curation; Software; Formal analysis; Investigation; Methodology. **George J Lu**: Conceptualization; Resources; Supervision; Funding acquisition; Writing—original draft; Project administration; Writing—review and editing.

Source data underlying figure panels in this paper may have individual authorship assigned. Where available, figure panel/source data authorship is listed in the following database record: biostudies:S-SCDT-10_1038-S44318-024-00178-2.

## Disclosure and competing interests statement

The authors declare no competing interests.

