## [Peer Review File · The EMBO Journal]

Elucidating the assembly of gas vesicles by systematic protein-protein interaction analysis

Manuel Iburg, Andrew Anderson, Vivian Wong, Erica Anton, Art He, and George Lu

Corresponding author(s): George Lu (george.lu@rice.edu)

Review Timeline:

Submission Date:	4th Oct 23
Editorial Decision:	24th Dec 23
Revision Received:	22nd Mar 24
Editorial Decision:	7th May 24
Revision Received:	31st May 24
Accepted:	19th Jun 24

Editor: *Cornelius Schneider*

Transaction Report:

Dear Dr. Lu,

Thank you for submitting your manuscript for consideration by the EMBO Journal. The manuscript has been seen by three referees whose comments can be seen below.

As can be seen from the reports, all three referees found the results of high importance and interest and agree that the experimental strategy was innovative but also have several technical concerns as well as suggestions how to make the manuscript easier to follow. In general, I find the referee reports fair and productive and I would therefore to invite you to submit a revised version of the manuscript, addressing the comments of all three reviewers. I should add that it is EMBO Journal policy to allow only a single round of revision, and acceptance of your manuscript will therefore depend on the completeness of your responses in this revised version. If you have any additional questions or want to discuss the revisions further, I am happy to do so by email or video conferencing.

We generally allow three months as standard revision time, which can be extended to 6 months in case of major revisions. As a matter of policy, competing manuscripts published during this period will not negatively impact on our assessment of the conceptual advance presented by your study. However, we request that you contact the editor as soon as possible upon publication of any related work, to discuss how to proceed. Should you foresee a problem in meeting the deadline, please let us know in advance and we may be able to grant an extension.

Thank you for the opportunity to consider your work for publication. I look forward to your revision.

Yours sincerely,

Cornelius Schneider

Cornelius Schneider, PhD
Editor
The EMBO Journal
c.schneider@embojournal.org

We realize that it is difficult to revise to a specific deadline. In the interest of protecting the conceptual advance provided by the work, we recommend a revision within 3 months (23rd Mar 2024). Please discuss the revision progress ahead of this time with the editor if you require more time to complete the revisions. Use the link below to submit your revision:

Referee #1:

The ambitious study of the interactions within 11 gas vesicle proteins designed, built, and tested 127 plasmids. This study ultimately categorized 124/1008 protein-protein interactions as potentially relevant to gas vesicle assembly. The scale and logic of the study were innovative to understand more about the roles of proteins in gas vesicle assembly. Based upon these protein-protein interactions and their dependencies, the authors proposed a model for the role of proteins in GV assembly, laying the foundation for future studies to interrogate the proposed model. This study should interest the synthetic biology and GV assembly communities.

I encourage the authors to consider the following major concerns:

1. The approach to tracking temporal changes in protein-protein interactions is an interesting one. This is a crucial experiment to dissect models of gas vesicle assembly as times were picked for when they first appeared and when they were assembled robustly. However, a caveat of this approach is that other biochemical processes may exhibit an effect when the time is from 4 to 24 hours. For example, the authors included one excellent control - western blot analysis. Inspection of Figure S2 of Lg-K, Lg-T, and Lg-B display significantly increased proteolysis at 24 hrs versus 4 hrs. In addition, Lg-N, Lg-S, Lg-J, L-Lg displayed some proteolysis increase after 24 hrs. These truncated products may result in synthetic reduced or increased binding, as opposed to differences in assembly mechanism.

a. To avoid this caveat, this analysis may point to more reliable interactions between full-length proteins at 4 hrs. vs 24 hrs.

b. The caption also states, "We sampled a subset of the samples" by western blot. It's unclear to the reviewer which samples were sampled and which were not. Those samples that were not sampled may run the risk of a higher and unknown level of proteolysis. Since proteolysis was observed in some samples, ideally, all samples should be analyzed by western blot. However, at the very least, a table of samples analyzed and those that were not should be made clear. This may complicate interpretations of changes in protein-protein interactions that are displayed.

c. Additionally, whether the cells are in an exponential or stationary phase may affect assembly somehow. At 4 and 24 hrs., what is the O.D. of the cells? Are both cell populations in the stationary phase?

2. Inferring that interactions are direct or indirect based on the removal of proteins is an interesting idea. A more precise interpretation is that this assay reports on the interaction dependencies. The interpretation of outright direct interactions may be accurate but may risk being over-interpreted. Do the design experiments cover all scenarios in which a protein-protein interaction is mediated by an auxiliary factor that may be redundant? For example, the observed interaction between proteins A & B requires C, D, or E in other proteins. Typically, the gold standard for direct interactions would be an in vitro protein-protein interaction assay. Specifying the interaction dependencies is a significant step forward in understanding GV assembly.

I encourage the authors to consider the following minor concerns:

3. In some cases, the authors make a hypothesis with little follow-up. For the following statements, could the author provide additional key references and arguments to support the foundation of their hypothesis? This would help readers assess the plausibility of the hypothesis.

a. "plausible to hypothesize that GvpF functions as a chaperone of GvpB."

b. "The chaperoning of GvpB, Stage III, likely occurs at the ribosome and in parallel to the initiation and elongation process of

the GV particles."

c. "We hypothesize that GvpN uses energy to prepare the correct folding of GvpB for insertion into the growing cylindrical portion of GVs."

Referee #2:

Summary and general comments:

The paper by Iburg and Anderson, et al. describes the development of a targeted and plasmid based in cellulo assay for classifying protein-protein interactions involved in gas vesicle (GV) formation. The first paper introduces the question/problem by providing a descriptive comparison of the physics of synthetic microbubbles versus the biophysics of natural and gene-encoded GVs. They go on to describe key 'knowledge gaps' regarding the formation of GVs, specifically regarding the molecular interactions that lead to their within cells. These descriptions set up the main motivation for the paper-the aim of understaining GV biogenesis by classifying the protein-protein interactions (PPIs) that underlie their formation in bacteria. The paper itself involves a high-/medium-throughput approach to classify such interactions. They use a plasmid based strategy to test combinations of 11 gene products encoded within a bacterial GV operon. A targeted screening strategy was devised using split Nano-Luciferase (NanoLuc) and screening was carried out in an orthogonal system, BL-21 E. coli-which do not express their own GVs, but with which gene expression can be tightly regulated using IPTG. Putative PPIs identified by NanoLuc activity were further validated, and an approach to identify templated interactions ternary interaction was also devised.

Overall, I think the paper is very interesting. Much current synthetic biology involves taking well-studied natural components and using them as biological parts to engineer cells. Here the authors take a reverse approach-in which GV gene components are investigated via orthogonal and IPTG-induced synthetic expression in BL21 E. coli. Whereas an aim of synbio is to make cells work like computers, here Lu and team want to learn why GVs work better than man-made synthetic microbubbles. Beyond conceptual innovations, the developed method also represents a valuable technical innovation and tool. The work lays a foundation for further dissection and understanding microbiological nanoassemblies within cells.

Aspects that should be addressed:

Whereas others have taken reductionist approaches dissecting biology, the authors here are applying a synthetic approach to elucidate nature's mechanisms. This is a very innovative approach in my opinion. My only real criticism is that the manuscript is somewhat overwhelming-this is a lot of information to consume. I have to say that it was hard for me to get through.... and thus I think the manuscript would benefit from some streamlining.

In its current form, the paper spends a lot of time explaining how the assay was set up. This is great, but I believe these descriptions should be in service of both validating the technique and setting up the reader to understand what was found. Are there components of the first few Results subsections that can be relegated to the Supplement? One thing that might help is to edit the Results subheadings to be declarative instead of descriptive... and then working to make sure that subsection speaks to its associated declaration. For example, one subheading might be: "X templates a n indirect interaction with Y and Z"

Also, this might be my own personal preference, but here is my take on the Introduction and Discussion sections. The last paragraph(s) of the Intro and the first one(s) of the Discussion should give readers the "jist" of the paper. At the end of the Intro, it is important for readers to fully appreciate such jist before diving into the Results. This will help readers better appreciate the work and it will also help reviewers better scrutinize the findings.

Minor comments:

- On Page 1, paragraph 3: here first use of the term " sodium dodecyl sulfate" should be followed with a definition of the abbreviation "(SDS)" which is used later in the text.

- Figure 2 should be broken down further into additional panels. This is especially important to do for panel B. It should be very clear to the reader what is hypothetical vs what is real data. For example, the schematic in Figure 2 B-II representing luminescence measurements is clearly a schematic, but I think it would be worth making that more immediately clear.... Perhaps a cartoon with some tubes with emitted photons etc, instead of a plot? I appreciate the "N= 3" as it reflects rigorous experimental planning... but it is more important to place such a designation in the panel below. Overall, the data should have their own subpanel letters and the usual information needed to interpret such data should be included there. For example... in the western blot, which antibody?

- kD should be kDa when referring to protein mass ("kilo-Dalton"). kD/Kd refers to an equilibrium binding constant.

- Much content through the Results text and Figure captions can be moved to methods and/or supplementary material. Example, last two sentences of caption for Figure 4E.

- Where they say "Next, we chose the split luciferase complementation assay to determine their interactions because of the high dynamic range, low background, commercially available substrates, and compatibility with experiments in living cells^{38,39}" ... It would be worth noting here directly that the assay is based on split-NanoLuc.

- The last paragraph starting on Page 2, and running into Page 3, which describes the calculations used to determine interaction possibilities, is a lot to consume. It could be useful to depict this in a figure as a workflow.

- The ribosome schematic in Figure 7-III depicts GvpF and GvpB as being thread through the interface that lies between large and small ribosomal subunits... but this is normally where mRNAs are thread through. A nascent protein would be more accurately depicted as emerging from the polypeptide exit tunnel, which should lie above the mRNA and exist within only the large subunit.

Referee #3:

In this study, the authors design a three-plasmid system based on a split luciferase assay to interrogate the pair-wise interaction network of the proteins encoded by a polycistronic gene cluster from *B. megaterium* required for the assembly of microbial gas vesicles. While the principal components contained in mature gas vesicles are known and well characterised, the function of most other proteins encoded by the gene cluster, and their temporal requirement during the gas vesicle assembly process remain largely unknown. The three-plasmid system provides an effective way to simplify screening the large number of combinatorial pairings and provides a useful tool set for more focused studies on subnetworks in the future. The authors not only comprehensively measure interaction strengths but also evaluate whether fusion modifications affect the ability of gas vesicles to form. The study complements earlier attempts to probe gas vesicle protein interactions in a halophilic archeon *H. salinarum*, which used split-GFP assay (Winter 2018, Völkner 2020, Jost 2022). The present study goes significantly beyond these earlier reports in systematically probing the effect of C- and N-terminal tag location and the interdependence of a particular interaction on other gas vesicle proteins by using background expression of the unmodified gene cluster, or specific deletion constructs and to perform the analysis at two different time points. This leads the authors to identify several relevant interactions, both confirming earlier data but also refining earlier models and discovering and providing testable hypotheses for new subnetworks. Even though the high-throughput assay alone does not provide conclusive evidence for proposed functions of gene products, the study and the datasets it has generated will be invaluable for guiding future experiments focused on understanding the biology of gas vesicle nucleation and growth.

Specific comments:

- Several previous studies have also investigated pair-wise interactions of gas vesicle proteins using split-GFP (Winter 2018, Völkner 2020, Jost 2022). While the authors mention and relate to these studies in the results section, it would be appropriate to mention these data and the current state of knowledge in the introduction. The authors should explain what motivated them to further investigate these interactions, how their approach differs and how it may be more powerful (e.g. inclusion of background proteins).

- The major gas vesicle protein is called GvpA in nearly all studies since the identification of the protein (de Marsac 1985, Hayes 1986). Multiple copies of GvpA from gene duplication events as they occur in several gene clusters are typically referred to as GvpA1, GvpA2, The only occurrence of the GvpB terminology in the literature is in the operon from *Bacillus megaterium* (Li 1998), and this terminology is confusing. For consistence, the nomenclature followed by UniProt is GvpA1, (A2, ...) and hence *B. megaterium* GvpB has recently been renamed to GvpA2 (entry O68677 · GVPA2_PRIMG). We suggest the authors consider following this convention, which will also help the clarity in the manuscript.

- Figure 1:

o A/B: For Figure 1 A the dimension of the scale bar should be given. Related to Figure 1B, the average diameter of gas vesicles from this operon is ~52 nm (Dutka 2021) not 70 nm.

o D: the use of blue and orange is confusing given the equivalent colour choice (with different meaning) for GvpA2/B and other gene cluster members in Figure 1C. Please consider adapting the colour scheme in either sub-panel. Using the same scheme for the POI in Figure 1E would further clarify the figure.

o D: for clarity, it would be useful for a broad readership to schematically introduce how the split-luciferase assay leads to detectable signal.

- Page 4: Interaction signal strengths were translated into the terms 'strong', 'significant', 'notable'. While this approach can help in interpretation of the results, the term 'significant' is problematic as it has well-established meaning in statistics, which is not the

way it is used here. The authors should consider finding a different term. The authors should also provide a rationale for the chosen thresholding in their classification - if there is literature categorising observed interaction signals in split-luciferase assays this should be referenced.

- On the note of significance: the authors state that their interaction measurements have been done in triplicate biological replicates, but statistical testing appears to be entirely absent throughout the manuscript (this holds for data presented in Figure 3C,F and Figure 4C,D as well as Figure 5A,C and Figure 6A). This analysis should be performed, and standard deviations and p-values should be reported to assess the statistical significance of the findings. For example, to test whether an observed interaction is significant, hypothesis testing against the negative control should be performed. Likewise, the statistical significance of the difference between the two timepoints of the temporal analysis should be determined.

- On page 10 the authors state: "GvpJ and GvpS may behave differently from GvpB and have less exposed hydrophobic surface". This speculation seems unfounded given the highly similar number of hydrophobic amino acids and very high homology mapped to their secondary structure (Figure 4E).

- In the discussion, some statements are unnecessarily speculative. Some examples are given below:

o "The chaperoning of GvpB, Stage III, likely occurs at the ribosome and in parallel to the initiation and elongation process of the GV particles." While it is indeed likely that chaperoning starts con-translationally, nothing in the presented data argues for or against this scenario.

o "We propose that GvpF and GvpG are the chaperones of GvpB and prevent it from becoming thermodynamically trapped and forming proteotoxic amyloid species." There is no evidence that GvpB forms amyloid species upon misfolding.

- Uncropped western blots should be shown in the supplement.

- The study has generated a comprehensive set of plasmids with broad utility for the field. We encourage the authors to make their plasmid sets publicly available.

Arjen Jakobi and Stefan Huber

We sincerely appreciate all the reviewers for their insightful feedback and enthusiasm. Our responses are detailed below in blue font. Additionally, the revised sections in the manuscript are highlighted in blue for clarity.

Referee #1:

The ambitious study of the interactions within 11 gas vesicle proteins designed, built, and tested 127 plasmids. This study ultimately categorized 124/1008 protein-protein interactions as potentially relevant to gas vesicle assembly. The scale and logic of the study were innovative to understand more about the roles of proteins in gas vesicle assembly. Based upon these protein-protein interactions and their dependencies, the authors proposed a model for the role of proteins in GV assembly, laying the foundation for future studies to interrogate the proposed model. This study should interest the synthetic biology and GV assembly communities.

I encourage the authors to consider the following major concerns:

1. The approach to tracking temporal changes in protein-protein interactions is an interesting one. This is a crucial experiment to dissect models of gas vesicle assembly as times were picked for when they first appeared and when they were assembled robustly. However, a caveat of this approach is that other biochemical processes may exhibit an effect when the time is from 4 to 24 hours. For example, the authors included one excellent control - western blot analysis. Inspection of Figure S2 of Lg-K, Lg-T, and Lg-B display significantly increased proteolysis at 24 hrs versus 4 hrs. In addition, Lg-N, Lg-S, Lg-J, L-Lg displayed some proteolysis increase after 24 hrs. These truncated products may result in synthetic reduced or increased binding, as opposed to differences in assembly mechanism.

We greatly appreciate the comments and suggestions provided. We agree that the transgene expression setup and the introduction of fusion proteins may introduce potential artifacts and unintended interactions into our dataset. In particular, the expression of transgenes over time can bring in competing effects that may confound the “base” interaction measures of two proteins of interest. During the design phase of this study, we were aware of the potential discrepancy that could arise between the t_4 and t_{24} datasets. However, we made the decision to proceed and present all the datasets to the readers, believing that this comprehensive dataset would best serve future studies aimed at dissecting detailed interactions and protein complexes.

Specifically, regarding the change of protein level and proteolysis, we have taken into consideration of the following situations:

- A) Overall protein levels in the cytosol will increase over time by extending expression time, but this is also true for the positive / negative controls. Therefore, normalizing our data points against these controls should mitigate the influence of protein level on our results. Furthermore, we divide all datapoints by the OD600 of the tested bacterial culture to normalize for total cell mass. Together, we anticipate that these two measures could mitigate the confounding effects produced by an increase in recombinant protein concentration inside the cells.

B) Regardless of timepoint, truncated transcripts or cryptic start codons within the coding sequence (CDS) may lead to incomplete peptides being synthesized, which may alter interaction dynamics. However, these effects likely occur even in the “native-like” GV operon and could therefore be relevant to the question of GV assembly.

C) Accumulation of recombinant proteins over time may lead to increased proteolysis or aggregation, changing the interaction dynamics of proteins or removing them from the pool of interacting species. Because our LgBit and SmBit test constructs express a single CDS as opposed to the polycistronic GV operon, this effect likely introduces artifacts into our measurement data. Nonetheless, our western blotting data in Figure 3 demonstrates that for all samples, the full-length LgBit fusion protein is the major species, so that the “real” interaction likely represents the majority of the signal.

In response to these comments and thoughts, we have added passages in our results and discussion sections to elaborate on and to emphasize the semi-quantitative nature of our dataset (Page 6, lines 271-277 & page 8, lines 382-385 & lines 388-390).

a. To avoid this caveat, this analysis may point to more reliable interactions between full-length proteins at 4 hrs. vs 24 hrs.

Thank you for this suggestion. We are mindful of these effects when analyzing our data, and below are two specific examples:

(1) As you pointed out, the Lg-L fusion protein shows a high degree of truncated species at t_{24} vs t_4 (Supplemental Figure 2, bottom left). Importantly, the increase over time in the interaction with GvpB (Figure 5 C&D), which is our “showpiece” change over time, was measured using the L-Lg plasmid in which the truncation effect at t_{24} is absent (Supplemental Figure 2, center right) and we would like to maintain this measurement as an example of our change over time hypothesis.

(2) Additionally, we do not attribute the additional signal for Lg-B, B-Lg (Supplemental Figure 2, bottom right) to proteolysis or truncation. As most of the additional bands are larger than the expected fusion protein (split NanoLuc LgBit + GvpB approximately 25 kDa), it is likely that we are observing SDS-insoluble oligomers of the major shell protein, forming a ladder of multimers and a high molecular weight smear. This effect is also visible to a lesser degree for Lg-J.

In addition to the passage in the results section mentioning this effect (Page 6, lines 271-277), we also added a passage to the discussion to address this (Page 8, lines 382-385 & 388-390).

b. The caption also states, "We sampled a subset of the samples" by western blot. It's unclear to the reviewer which samples were sampled and which were not. Those samples that were not sampled may run the risk of a higher and unknown level of proteolysis. Since proteolysis was observed in some samples, ideally, all samples should be analyzed by western blot. However, at the very least, a table of samples analyzed and those that were not should be made clear. This may complicate interpretations of changes in protein-protein interactions that are displayed.

Thank you for this suggestion. We agree that differences in the integrity of the fusion proteins over time could affect the outcome of measurements, especially at t_{24} . We made the following considerations when deciding how many samples to take for western blotting:

- A) To sample all the measurement points (approximately 3000), we would have to conduct about 200 western blot experiments, which is practically challenging and will massively expand the timeline of the project.
- B) We sample a subset of all measurements for western blotting under the assumption that the expression and stability of any given fusion protein will be consistent across the $n = 3$ repeats, e.g. B-Lg are representative of all instances of B-Lg.
- C) Considering the total workload of experimentation and aiming to cover the samples broadly, we sampled at least 3 independent repeats of each Gvp-Lg-FLAG or FLAG-Lg-Gvp at both timepoints t_4 and t_{24} .
- D) Our initial intention in performing western blot experiments alongside the interaction measurements was to confirm that for any given negative measurement, the absence of signal is not due to the absence of protein expression, and we believe this goal has been satisfied by the western blot experiments. It is conceivable that if we had a blot to match every measurement point, we could attempt relative or absolute quantification to normalize all interaction data to the relative protein expression levels. However, even in this scenario we would expect proteolysis or aggregation and truncated peptides to alter the protein interaction dynamics, rendering us unable to present an “absolute” interaction value for each pair of fusion proteins.

To address this concern, we have added a column in Appendix Table 2 to indicate which of the respective measurements have been subjected to Western blotting analysis.

c. Additionally, whether the cells are in an exponential or stationary phase may affect assembly somehow. At 4 and 24 hrs., what is the O.D. of the cells? Are both cell populations in the stationary phase?

Thank you for pointing out this effect of *E. coli* growth phase on our data. In brief, most cultures measured at t_4 were between OD₆₀₀ 0.8 to 1.0, and thus shortly past steady state growth, while most cultures at t_{24} had reached a higher cell density (between OD₆₀₀ 1,0 to 1,5) but not yet exhausted the media (we consulted Sezonov et al., 2007 for considerations on the relation between OD₆₀₀ and growth stage (Sezonov *et al*, 2007)). More exhaustively, we would like to bring up the following considerations:

- A) Heterologous expression of gas vesicles is non-physiological for *E. coli* and introduces a high metabolic burden, particularly due to using a set of three plasmids and three antibiotics. We also now document the bulk OD₆₀₀ of all samples in Appendix Figure S5A. The growth rate of our assay cultures is different from standard *E. coli* growth rates (please also refer to the bulk OD₆₀₀ dataset we provide in Appendix Figure S5A). In fact, in a given population of *E. coli* heterologously expressing GVs, after harvesting cells a fraction of the population has undergone cell lysis (as observed by free-floating GVs; data not shown).
- B) We designed our experiments to emulate the process of producing GVs in *E. coli* as closely as possible to mimic GV assembly as it would occur in application scenarios e.g. for diagnostics (cited in the introduction and discussion sections of the manuscript). Consequently, we can observe the assembly of GVs in multiple of our measurements (Figure 6) and

we take this to indicate that our measurements represent the actual conditions of GV assembly. Notably, since GV assembly in application scenarios would also introduce a metabolic burden, we consider these conditions as relevant.

C) In anticipation of your next concern, the metabolic strain introduced by GV expression may also cause a stress response in *E. coli*, such as the heat shock response. Hypothetically, upregulation of molecular chaperones in *E. coli* may play a role in GV assembly (Jung *et al*, 2021), introducing the possibility that endogenous proteins of the host cell are involved in GV assembly, posing a new group of “unknown third” interaction partners. To limit the scope of this study to a reasonable level, we did not investigate the role of protein interaction partners outside of the GV operon discussed.

In our manuscript, to address this concern, we mention the growth phase of our *E. coli* cells at time points t_4 and t_{24} in our results section and added Appendix Figure S5 giving a bulk overview of the OD₆₀₀ values at the respective timepoints. In brief, the cell density is significantly different at both time points, but both groups are beyond steady state growth and have not yet exhausted their media.

2. Inferring that interactions are direct or indirect based on the removal of proteins is an interesting idea. A more precise interpretation is that this assay reports on the interaction dependencies. The interpretation of outright direct interactions may be accurate but may risk being over-interpreted. Do the design experiments cover all scenarios in which a protein-protein interaction is mediated by an auxiliary factor that may be redundant? For example, the observed interaction between proteins A & B requires C, D, or E in other proteins. Typically, the gold standard for direct interactions would be an *in vitro* protein-protein interaction assay. Specifying the interaction dependencies is a significant step forward in understanding GV assembly.

Thank you for the comments and the encouragement of pursuing interaction dependencies. We agree that, ideally, we would exhaustively delete individual Gvps and combinations of GV proteins to assess their influence on a given pair of protein-protein interactions. However, such an approach would significantly increase the number of experiments required, rendering it practically infeasible. To investigate the complete set of all dependent protein interactions for an operon with 11 proteins, we would need to conduct $5368 * 3 = 16104$ repeat experiments, excluding positive and negative controls (refer to the newly added Appendix Figure S5B). Consequently, in the present study, we have opted to probe only a few identified protein subnetworks, and we selectively deleted proteins already known to have connections with the primary interaction pair we sought to probe. Regarding the *in vitro* protein-protein interaction assay, thank you for bringing this up. It is useful to note that not all GV proteins have established purification protocols at this moment. We are actively pursuing research to obtain purified GV proteins and hope to conduct *in vitro* reconstitution experiments of GV protein complexes and the GV assembly process in the near future.

To address the comments, we have expanded our discussion to explain that our study of interaction dependency is based on a subtractive method and could possibly miss interactions dependent on native *E. coli* proteins and OR-type interaction dependencies (Page 4, lines 144-146 & page 8, lines 391-397). We also elaborated on our choice to perform our interaction study mostly *in vivo* in the Appendix results section. Furthermore, we changed our writing to reflect that this

method elucidates interaction dependencies, but not necessarily directness or indirectness of interactions. Appendix 5B, as well as the Appendix results section, contain a more in-depth consideration about the scale of experiments.

Additionally, below we have provided further background information on our thought process during the planning of these experiments. We have noted that the datasets presented in this manuscript would have different levels of completeness (as displayed in the different sheets of Appendix Table S2):

A) In Figure 3, our goal was to provide a complete dataset of all A to B interactions between the 11 proteins in the GV operon and if an interaction was not observed, we varied the geometry of the fusion proteins (C-terminal, N-terminal, C & N-terminal) until we either confirmed an interaction or confirmed that no interaction was measured in any configuration.

As you pointed out, in these cases the interaction measured could be dependent on other proteins of the GV operon (which we addressed in Figure 4) or even on “third party” proteins which are natively present in *E. coli* (and presumably homologous to proteins found in *B. megaterium*, such as molecular chaperones). An observation of native proteins being involved in GV formation has also been made for *S. cerevisiae* (Jung *et al.*, 2021). We considered the latter case to be beyond the scope of this study.

B) To address co-dependency of interactions within the GV operon, for every interaction detected in our first screening, we subsequently repeated the interaction experiment with the same two fusion proteins, but in the absence of the 9 remaining members of the GV operon. For a fraction of the interactions, we then observed that they were diminished in this scenario, indicating that a dependency existed (e.g. on proteins C, D or E as you mentioned). Indeed, this method of determining dependencies does not confirm if the additional interaction partners are redundant or exchangeable (such as an OR dependency) - our screening for this part of the study was only subtractive. For another fraction of the interactions, we still observed the same interaction as before, indicating that either no dependency existed, or that a dependency existed on a protein natively present in *E. coli*. We concede that the latter case could confound some of the measurements made in this study but is not immediately relevant to the interactions of the GV operon as presented here (which are sufficient by themselves to allow GV synthesis) and it would be hard to screen for these additional interaction partners within a feasible number of experiments for this study.

C) Finally, for those protein pairs where deletion of the GV operon diminished interaction, we forwent a systemic screen and hypothesized which of the other GV proteins the interaction was dependent on. Typically, we chose those GV proteins that both interaction partners previously had interacted with and then utilized a triple deletion (fusion proteins A and B, as well as putative interaction partner C being absent, but all other GV proteins present) to identify if a protein C was mediating the interaction of A & B. Particularly interesting results of this screening are presented in Figure 5 A-D, as well as in Supplemental Table 2. We did not perform a complete screening of all possible triple deletions in the GV operon, which would have massively increased the number of experiments needed.

D) In previous studies on the interaction network between GV proteins, a split-GFP *in vivo* screen was combined with a column-based pull-down for *in vitro* interaction measurements (Volkner *et al.*, 2020). This experiment is in principle suitable to isolate the interaction between two proteins and identify true primary interactions, excluding the possibility of unknown third-party proteins being required for the interaction. We chose to not add an *in vitro* component to our study for the following reasons: firstly, we prioritized completeness of screening over depth of screening. While we think it is possible to scale up and parallelize a column-based *in vitro* interaction screen to supplement our *in vivo* data, it would

have increased the workload and cost of this study significantly and added the need for an additional 200+ SDS-PAGE experiments (as mentioned for western blotting above). We limited ourselves to studying mostly *in vivo* samples. Secondly, we believe that the *in vivo* screening of GV assembly interactions is more representative of the biological reality that underlies GV formation, as *in vitro* assembly of GVs has not yet been achieved. Isolation of individual GV proteins after cell lysis followed by immobilization of proteins to a solid support also introduces possible artifacts such as denaturing, and for some of the 11 GV proteins isolation for *in vitro* studies may not be feasible. Moreover, co-purification of binding partners on column from the *E. coli* lysate presents a challenge, mirroring the challenge of interactions being measured in the presence of an unknown third protein.

I encourage the authors to consider the following minor concerns:

3. In some cases, the authors make a hypothesis with little follow-up. For the following statements, could the author provide additional key references and arguments to support the foundation of their hypothesis? This would help readers assess the plausibility of the hypothesis.

Thank you for calling out the need to further elaborate on our hypotheses and provide pertinent citations to support our claims. In response, we have included additional citations as outlined below.

a. "plausible to hypothesize that GvpF functions as a chaperone of GvpB."

Our basis for hypothesizing that GvpF is a chaperone to GvpB is as follows:

A) Our data presented in Supplemental Table 2 And Figure 5 A&B suggests that GvpF binds GvpB in the absence of all other GV proteins, making it a primary interaction partner, but also that the sole presence of GvpG is sufficient to diminish the binding of B/F, indicating that this interaction is transitory. Völkner *et al.*, 2020 (Volkner *et al.*, 2020) show analogously for the major shell protein to be a primary interaction partner of GvpF.

B) The hypothesis of GvpF being a chaperone to GvpB is not original to this manuscript and was previously presented, such as in a recent review paper (Pfeifer, 2022).

C) Xu *et al.*, 2014 suggest that GvpF is a structural protein of GVs (Xu *et al.*, 2014), although CryoEM analysis of GVs does not appear to support this hypothesis (Dutka *et al.*, 2023; Huber *et al.*, 2023). Notably, this publication presents GvpF as binding GvpB on the *internal* surface of GVs, hence binding the hydrophobic portion of GvpB.

The role of GvpF as a transitory binding partner, targeting the hydrophobic portion of GvpB would be consistent with the role of a molecular chaperone (Hartl *et al.*, 2011) and leads us to hypothesize – not conclusively prove – that GvpF is a molecular chaperone to GvpB.

We have edited our manuscript to give a more complete argument for the role of GvpF (page 7, lines 330-339) and tuned down our claims about GvpF being a molecular chaperone to GvpB.

b. "The chaperoning of GvpB, Stage III, likely occurs at the ribosome and in parallel to the initiation and elongation process of the GV particles."

Based on previous findings that overexpression of the GV major shell protein can be proteotoxic (Jung *et al.*, 2021) and is partially rescued by overexpression of molecular chaperones, we consider it plausible that chaperoning of GvpB is required. This study shows that GvpF is the primary interaction partner of GvpB (in the absence of the other 9 GV proteins), which is in agreement with a previous study (Volkner *et al.*, 2020), where GvpF is also identified as the primary interaction partner of the major shell protein. This study identified the N-terminus of the major shell protein as the likely binding site of GvpF, and by analogy we propose that the binding of GvpF may occur cotranslationally. Regarding the statement that the chaperoning of GvpB occurs in parallel to the initiation and elongation process of the GV particles, we wrote this statement to underscore the likelihood that chaperoning takes place at a distinct site within the cytosol from where the GV assembly occurs.

In the main text of the article, we have tuned down our wording in the discussion to make clear that the cotranslational binding of GvpB and GvpF is plausible, but not confirmed (page 7, lines 332-334).

c. "We hypothesize that GvpN uses energy to prepare the correct folding of GvpB for insertion into the growing cylindrical portion of GVs."

As GVs stabilize a bubble of gaseous air, derived from dissolved gas, inside the cytosol, which is a decrease of entropy, we propose that both formation and growth of GVs require energy input. Outside of the energy utilized by the ribosome to synthesize GV proteins, GvpN is the only GV protein identified as an ATPase (of the AAA+ family), based on homology. Notably, Cai *et al.*, 2020 demonstrate that a purified homolog of GvpN (from *Anabaena* sp. PCC 7120) exhibits ATPase activity *in vitro* and we conclude by homology that the AAA+ domain of GvpN is active (Cai *et al.*, 2020). Deletion of GvpN from the operon does not abrogate GV formation but inhibits elongation and leads to the formation of shorter GVs, indicating that the energy is used by GvpN to elongate GVs (Pfeifer, 2022). The elongation of GVs likely occurs at the center of the cylinder and since the main component is GvpB (Huber *et al.*, 2023), and thus we hypothesize that GvpN utilizes energy to facilitate the insertion of GvpB into the growing portion of the cylinder.

We have expanded our manuscript to include additional literature references and explanations to support our claims (page 7, lines 349-354).

References

- Bayro MJ, Daviso E, Belenky M, Griffin RG, Herzfeld J (2012) An amyloid organelle, solid-state NMR evidence for cross-beta assembly of gas vesicles. *J Biol Chem* 287: 3479-3484
- Cai K, Xu BY, Jiang YL, Wang Y, Chen Y, Zhou CZ, Li Q (2020) The model cyanobacteria *Anabaena* sp. PCC 7120 possess an intact but partially degenerated gene cluster encoding gas vesicles. *BMC Microbiol* 20: 110

- Dutka P, Metskas LA, Hurt RC, Salahshoor H, Wang TY, Malounda D, Lu GJ, Chou TF, Shapiro MG, Jensen GJ (2023) Structure of *Anabaena flos-aquae* gas vesicles revealed by cryo-ET. *Structure* 31: 518-528 e516
- Hartl FU, Bracher A, Hayer-Hartl M (2011) Molecular chaperones in protein folding and proteostasis. *Nature* 475: 324-332
- Huber ST, Terwiel D, Evers WH, Maresca D, Jakobi AJ (2023) Cryo-EM structure of gas vesicles for buoyancy-controlled motility. *Cell* 186: 975-986 e913
- Jung H, Ling H, Tan YQ, Chua NH, Yew WS, Chang MW (2021) Heterologous expression of cyanobacterial gas vesicle proteins in *Saccharomyces cerevisiae*. *Biotechnol J* 16: e2100059
- Pfeifer F (2022) Recent Advances in the Study of Gas Vesicle Proteins and Application of Gas Vesicles in Biomedical Research. *Life (Basel)* 12
- Sezonov G, Joseleau-Petit D, D'Ari R (2007) *Escherichia coli* physiology in Luria-Bertani broth. *J Bacteriol* 189: 8746-8749
- Volkner K, Jost A, Pfeifer F (2020) Accessory Gvp Proteins Form a Complex During Gas Vesicle Formation of Haloarchaea. *Front Microbiol* 11: 610179
- Xu BY, Dai YN, Zhou K, Liu YT, Sun Q, Ren YM, Chen Y, Zhou CZ (2014) Structure of the gas vesicle protein GvpF from the cyanobacterium *Microcystis aeruginosa*. *Acta Crystallogr D Biol Crystallogr* 70: 3013-3022

Referee #2:

Summary and general comments:

The paper by Iburg and Anderson, et al. describes the development of a targeted and plasmid based in cellulo assay for classifying protein-protein interactions involved in gas vesicle (GV) formation. The first paper introduces the question/problem by providing a descriptive comparison of the physics of synthetic microbubbles versus the biophysics of natural and gene-encoded GVs. They go on to describe key 'knowledge gaps' regarding the formation of GVs, specifically regarding the molecular interactions that lead to their within cells. These descriptions set up the main motivation for the paper-the aim of understaining GV biogenesis by classifying the protein-protein interactions (PPIs) that underlie their formation in bacteria. The paper itself involves a high-/medium-throughput approach to classify such interactions. They use a plasmid based strategy to test combinations of 11 gene products encoded within a bacterial GV operon. A targeted screening strategy was devised using split Nano-Luciferase (NanoLuc) and screening was carried out in an orthogonal system, BL-21 E. coli-which do not express their own GVs, but with which gene expression can be tightly regulated using IPTG. Putative PPIs identified by NanoLuc activity were further validated, and an approach to identify templated interactions ternary interaction was also devised.

Overall, I think the paper is very interesting. Much current synthetic biology involves taking well-studied natural components and using them as biological parts to engineer cells. Here the authors take a reverse approach-in which GV gene components are investigated via orthogonal and IPTG-induced synthetic expression in BL21 E coli. Whereas an aim of synbio is to make cells work like computers, here Lu and team want to learn why GVs work better than man-made synthetic microbubbles. Beyond conceptual innovations, the developed method also represents a valuable technical innovation and tool. The work lays a foundation for further dissection and understanding microbiological nanoassemblies within cells.

We appreciate the reviewer's summary of the significance and impact of the work and the encouragement of our study.

Aspects that should be addressed:

Whereas others have taken reductionist approaches dissecting biology, the authors here are applying a synthetic approach to elucidate nature's mechanisms. This is a very innovative approach in my opinion. My only real criticism is that the manuscript is somewhat overwhelming-this is a lot of information to consume. I have to say that it was hard for me to get through.... and thus I think the manuscript would benefit from some streamlining.

In its current form, the paper spends a lot of time explaining how the assay was set up. This is great, but I believe these descriptions should be in service of both validating the technique and setting up the reader to understand what was found. Are there components of the first few Results subsections that can be relegated to the Supplement? One thing that might help is to edit the Results subheadings to be declarative instead of descriptive... and then working to make sure that

subsection speaks to its associated declaration. For example, one subheading might be: "X templates a n indirect interaction with Y and Z"

Thank you for the helpful editing suggestions! We have addressed this by condensing the technical description of our setup and moving part of it to the Appendix or the Methods section. We have also reviewed our subheadings, included more subheadings, and gave them more descriptive titles.

Also, this might be my own personal preference, but here is my take on the Introduction and Discussion sections. The last paragraph(s) of the Intro and the first one(s) of the Discussion should give readers the "jist" of the paper. At the end of the Intro, it is important for readers to fully appreciate such jist before diving into the Results. This will help readers better appreciate the work and it will also help reviewers better scrutinize the findings.

This is an excellent suggestion. We have edited our manuscript to make sure that the introduction and discussion end / start with a strong summary of our overall purpose.

Minor comments:

- On Page 1, paragraph 3: here first use of the term " sodium dodecyl sulfate" should be followed with a definition of the abbreviation "(SDS)" which is used later in the text.

Thank you for catching this. We have edited the text.

- Figure 2 should be broken down further into additional panels. This is especially important to do for panel B. It should be very clear to the reader what is hypothetical vs what is real data. For example, the schematic in Figure 2 B-II representing luminescence measurements is clearly a schematic, but I think it would be worth making that more immediately clear.... Perhaps a cartoon with some tubes with emitted photons etc, instead of a plot? I appreciate the "N= 3" as it reflects rigorous experimental planning... but it is more important to place such a designation in the panel below. Overall, the data should have their own subpanel letters and the usual information needed to interpret such data should be included there. For example... in the western blot, which antibody?

Thank you for sharing your analysis of our Figure 2. We have redesigned the figure with additional annotations, clearer subdivisions, and a more complete figure legend to clearly follow the purpose of introducing our experimental setup and be less confusing. Additional explanations have been given to distinguish cartoons and example data. Following EMBO J author guidelines, we included the antibodies used in the Methods section and the Author checklist.

- kD should be kDa when referring to protein mass ("kilo-Dalton"). kD/Kd refers to an equilibrium binding constant.

Thank you for catching this. We have made that change in our figures.

- Much content through the Results text and Figure captions can be moved to methods and/or supplementary material. Example, last two sentences of caption for Figure 4E.

We have addressed this by moving additional parts of our technical description to the Methods section or the Appendix. We decided to maintain some degree of explanation in the results section as we consider the experimental setup to be complex and the setup directly links to the results discussed.

- Where they say "Next, we chose the split luciferase complementation assay to determine their interactions because of the high dynamic range, low background, commercially available substrates, and compatibility with experiments in living cells^{38,39}" ... It would be worth noting here directly that the assay is based on split-NanoLuc.

Thanks for pointing this out. We have clarified the exact source of the split-NanoLuc we utilized.

- The last paragraph starting on Page 2, and running into Page 3, which describes the calculations used to determine interaction possibilities, is a lot to consume. It could be useful to depict this in a figure as a workflow.

Thank you for mentioning the issues with this paragraph. We have created a sketch (Supplementary Figure 5B) to illustrate calculations of the possible interactions to be analyzed in this study. We also cleared up our results section, moving parts of the exhaustive description to the Appendix.

- The ribosome schematic in Figure 7-III depicts GvpF and GvpB as being thread through the interface that lies between large and small ribosomal subunits... but this is normally where mRNAs are thread through. A nascent protein would be more accurately depicted as emerging from the polypeptide exit tunnel, which should lie above the mRNA and exist within only the large subunit.

Thank you for catching this. We apologize for the crude ribosome sketch and have updated Figure 7 to depict the nascent GV proteins exiting the polypeptide exit tunnel more accurately.

References

- Bayro MJ, Daviso E, Belenky M, Griffin RG, Herzfeld J (2012) An amyloid organelle, solid-state NMR evidence for cross-beta assembly of gas vesicles. *J Biol Chem* 287: 3479-3484
- Cai K, Xu BY, Jiang YL, Wang Y, Chen Y, Zhou CZ, Li Q (2020) The model cyanobacteria *Anabaena* sp. PCC 7120 possess an intact but partially degenerated gene cluster encoding gas vesicles. *BMC Microbiol* 20: 110
- Dutka P, Metskas LA, Hurt RC, Salahshoor H, Wang TY, Malounda D, Lu GJ, Chou TF, Shapiro MG, Jensen GJ (2023) Structure of *Anabaena flos-aquae* gas vesicles revealed by cryo-ET. *Structure* 31: 518-528 e516

- Hartl FU, Bracher A, Hayer-Hartl M (2011) Molecular chaperones in protein folding and proteostasis. *Nature* 475: 324-332
- Huber ST, Terwiel D, Evers WH, Maresca D, Jakobi AJ (2023) Cryo-EM structure of gas vesicles for buoyancy-controlled motility. *Cell* 186: 975-986 e913
- Jung H, Ling H, Tan YQ, Chua NH, Yew WS, Chang MW (2021) Heterologous expression of cyanobacterial gas vesicle proteins in *Saccharomyces cerevisiae*. *Biotechnol J* 16: e2100059
- Pfeifer F (2022) Recent Advances in the Study of Gas Vesicle Proteins and Application of Gas Vesicles in Biomedical Research. *Life (Basel)* 12
- Sezonov G, Joseleau-Petit D, D'Ari R (2007) *Escherichia coli* physiology in Luria-Bertani broth. *J Bacteriol* 189: 8746-8749
- Volkner K, Jost A, Pfeifer F (2020) Accessory Gvp Proteins Form a Complex During Gas Vesicle Formation of Haloarchaea. *Front Microbiol* 11: 610179
- Xu BY, Dai YN, Zhou K, Liu YT, Sun Q, Ren YM, Chen Y, Zhou CZ (2014) Structure of the gas vesicle protein GvpF from the cyanobacterium *Microcystis aeruginosa*. *Acta Crystallogr D Biol Crystallogr* 70: 3013-3022

Referee #3:

In this study, the authors design a three-plasmid system based on a split luciferase assay to interrogate the pair-wise interaction network of the proteins encoded by a polycistronic gene cluster from *B. megaterium* required for the assembly of microbial gas vesicles. While the principal components contained in mature gas vesicles are known and well characterised, the function of most other proteins encoded by the gene cluster, and their temporal requirement during the gas vesicle assembly process remain largely unknown. The three-plasmid system provides an effective way to simplify screening the large number of combinatorial pairings and provides a useful tool set for more focused studies on subnetworks in the future. The authors not only comprehensively measure interaction strengths but also evaluate whether fusion modifications affect the ability of gas vesicles to form. The study complements earlier attempts to probe gas vesicle protein interactions in a halophilic archeon *H. salinarum*, which used split-GFP assay (Winter 2018, Völkner 2020, Jost 2022). The present study goes significantly beyond these earlier reports in systematically probing the effect of C- and N-terminal tag location and the interdependence of a particular interaction on other gas vesicle proteins by using background expression of the unmodified gene cluster, or specific deletion constructs and to perform the analysis at two different time points. This leads the authors to identify several relevant interactions, both confirming earlier data but also refining earlier models and discovering and providing testable hypotheses for new subnetworks. Even though the high-throughput assay alone does not provide conclusive evidence for proposed functions of gene products, the study and the datasets it has generated will be invaluable for guiding future experiments focused on understanding the biology of gas vesicle nucleation and growth.

Specific comments:

- Several previous studies have also investigated pair-wise interactions of gas vesicle proteins using split-GFP (Winter 2018, Völkner 2020, Jost 2022). While the authors mention and relate to these studies in the results section, it would be appropriate to mention these data and the current state of knowledge in the introduction. The authors should explain what motivated them to further investigate these interactions, how their approach differs and how it may be more powerful (e.g. inclusion of background proteins).

Thank you for pointing this out. We have expanded our introduction section to contextualize our work better and explain what we specifically add to the investigation (page 2, lines 45-52). We also added an extended discussion section to the Appendix to contextualize our research better (Appendix discussion section pages 14-17, lines 259-355).

- The major gas vesicle protein is called GvpA in nearly all studies since the identification of the protein (de Marsac 1985, Hayes 1986). Multiple copies of GvpA from gene duplication events as they occur in several gene clusters are typically referred to as GvpA1, GvpA2, The only occurrence of the GvpB terminology in the literature is in the operon from *Bacillus megaterium* (Li 1998), and this terminology is confusing. For consistence, the nomenclature followed by UniProt is GvpA1, (A2, ...) and hence *B. megaterium* GvpB has recently been renamed to GvpA2 (entry

O68677 · GVPA2_PRIMG). We suggest the authors consider following this convention, which will also help the clarity in the manuscript.

We are grateful for this clarification. For clarity, we edited our manuscript to use GvpA2 instead of GvpB (outside of this revision section). We have also addressed the naming of GvpB in our manuscript.

- Figure 1:

o A/B: For Figure 1 A the dimension of the scale bar should be given. Related to Figure 1B, the average diameter of gas vesicles from this operon is ~52 nm (Dutka 2021) not 70 nm.

o D: the use of blue and orange is confusing given the equivalent colour choice (with different meaning) for GvpA2/B and other gene cluster members in Figure 1C. Please consider adapting the colour scheme in either sub-panel. Using the same scheme for the POI in Figure 1E would further clarify the figure.

o D: for clarity, it would be useful for a broad readership to schematically introduce how the split-luciferase assay leads to detectable signal.

Thank you for the helpful suggestions. We have clarified the length of the scalebar and the dimensions of GVs in the sketch. We have also adjusted the color scheme of our figure and added a small sketch explaining the mechanics of split-luciferase complementation (Figure 1E).

- Page 4: Interaction signal strengths were translated into the terms 'strong', 'significant', 'notable'. While this approach can help in interpretation of the results, the term 'significant' is problematic as it has well-established meaning in statistics, which is not the way it is used here. The authors should consider finding a different term. The authors should also provide a rationale for the chosen thresholding in their classification - if there is literature categorising observed interaction signals in split-luciferase assays this should be referenced.

Thank you for the comment and we apologize for this oversight. We have changed the naming of our bins into high / medium / low. The rationale for choosing the bins as we did was arbitrary, with the goal of highlighting a limited number of outstanding interactions while not overcrowding our figures. We want to point out that for a more precise analysis, Appendix Table S2 has a less visual, more comprehensive dataset. We have also edited our manuscript to explain the reasoning behind setting the bins (page 3, lines 102-106).

- On the note of significance: the authors state that their interaction measurements have been done in triplicate biological replicates, but statistical testing appears to be entirely absent throughout the manuscript (this holds for data presented in Figure 3C,F and Figure 4C,D as well as Figure 5A,C and Figure 6A). This analysis should be performed, and standard deviations and p-values should be reported to assess the statistical significance of the findings. For example, to test whether an observed interaction is significant, hypothesis testing against the negative control should be performed. Likewise, the statistical significance of the difference between the two timepoints of the temporal analysis should be determined.

This is a very good point! To address this, we have performed t-tests (using the negative control as reference) and provided p-values for all measurements in Appendix Table S2. We also tested all t24 vs t4 measurements in this manner and indicated p-values in the table. Notably, we edited parts of our results and discussion section to exclude reported interactions that were not statistically significant, and this was reflected in Figures 3 and 4. The overall content and conclusions of this manuscript are not significantly changed by these alterations. We addressed the statistical testing in the results section and gave a more comprehensive explanation in the Methods section (pages 10-11, lines 514-528). We also clarified statistical testing in Figure 6A and addressed it in the figure legend (Figure legend 6).

- On page 10 the authors state: "GvpJ and GvpS may behave differently from GvpB and have less exposed hydrophobic surface". This speculation seems unfounded given the highly similar number of hydrophobic amino acids and very high homology mapped to their secondary structure (Figure 4E).

Thank you for pointing this out. This statement is indeed hypothetical, and we have removed these claims as to why GvpJ and GvpS show less pronounced proteotoxicity despite being sequence homologs to GvpB (page 5, lines 225-227).

- In the discussion, some statements are unnecessarily speculative. Some examples are given below:

o "The chaperoning of GvpB, Stage III, likely occurs at the ribosome and in parallel to the initiation and elongation process of the GV particles." While it is indeed likely that chaperoning starts con-translationally, nothing in the presented data argues for or against this scenario.

Thank you for this insight. We admit that the hypothesis is highly speculative and tuned down our statement in the discussion to avoid overinterpretation of existing data (page 7, lines 333-335).

o "We propose that GvpF and GvpG are the chaperones of GvpB and prevent it from becoming thermodynamically trapped and forming proteotoxic amyloid species." There is no evidence that GvpB forms amyloid species upon misfolding.

We apologize for formulating this statement in a misleading fashion. We were relying on the idea that the major shell protein of GVs may be a functional amyloid (Bayro *et al*, 2012), which may be supported by GvpA2 forming SDS-insoluble oligomers in our hands (Supplemental Figure 3, bottom right). Upon revisiting the data, we admit that further experimentation is required to position the major shell protein as amyloidogenic. However, recent studies of overexpression in yeast indicate that the major shell protein is proteotoxic unless chaperoned or introduced to a GV (Jung *et al.*, 2021). We have rewritten this section to make more conservative claims about proteotoxicity caused by GvpA2 overexpression (page 7, lines 331-333).

- Uncropped western blots should be shown in the supplement.

We have added an Appendix Figure S4 showing the uncropped original of all western blots presented. Cropped regions are indicated.

- The study has generated a comprehensive set of plasmids with broad utility for the field. We encourage the authors to make their plasmid sets publicly available.

We have initiated a deposition process with Addgene to make the plasmid set used here publicly available as Addgene deposit #83964. The submission is currently in preparation.

References

- Bayro MJ, Daviso E, Belenky M, Griffin RG, Herzfeld J (2012) An amyloid organelle, solid-state NMR evidence for cross-beta assembly of gas vesicles. *J Biol Chem* 287: 3479-3484
- Cai K, Xu BY, Jiang YL, Wang Y, Chen Y, Zhou CZ, Li Q (2020) The model cyanobacteria *Anabaena* sp. PCC 7120 possess an intact but partially degenerated gene cluster encoding gas vesicles. *BMC Microbiol* 20: 110
- Dutka P, Metskas LA, Hurt RC, Salahshoor H, Wang TY, Malounda D, Lu GJ, Chou TF, Shapiro MG, Jensen GJ (2023) Structure of *Anabaena flos-aquae* gas vesicles revealed by cryo-ET. *Structure* 31: 518-528 e516
- Hartl FU, Bracher A, Hayer-Hartl M (2011) Molecular chaperones in protein folding and proteostasis. *Nature* 475: 324-332
- Huber ST, Terwiel D, Evers WH, Maresca D, Jakobi AJ (2023) Cryo-EM structure of gas vesicles for buoyancy-controlled motility. *Cell* 186: 975-986 e913
- Jung H, Ling H, Tan YQ, Chua NH, Yew WS, Chang MW (2021) Heterologous expression of cyanobacterial gas vesicle proteins in *Saccharomyces cerevisiae*. *Biotechnol J* 16: e2100059
- Pfeifer F (2022) Recent Advances in the Study of Gas Vesicle Proteins and Application of Gas Vesicles in Biomedical Research. *Life (Basel)* 12
- Sezonov G, Joseleau-Petit D, D'Ari R (2007) *Escherichia coli* physiology in Luria-Bertani broth. *J Bacteriol* 189: 8746-8749
- Volkner K, Jost A, Pfeifer F (2020) Accessory Gvp Proteins Form a Complex During Gas Vesicle Formation of Haloarchaea. *Front Microbiol* 11: 610179
- Xu BY, Dai YN, Zhou K, Liu YT, Sun Q, Ren YM, Chen Y, Zhou CZ (2014) Structure of the gas vesicle protein GvpF from the cyanobacterium *Microcystis aeruginosa*. *Acta Crystallogr D Biol Crystallogr* 70: 3013-3022

Dear Dr Lu,

Thank you for submitting a revised version of your manuscript. Your study has now been seen by the original referees, who find that their previous concerns have been addressed and now recommend publication of the manuscript. There remain only a few mainly editorial points that have to be addressed before I can extend formal acceptance of the manuscript:

- FUNDING INFO: missing info in ms file - grant numbers not listed: 511048568; MF-2012-01314; C-2069-20210327; RR190081
- COI: title needs renaming to "DISCLOSURE AND COMPETING INTERESTS STATEMENT"
- AC/CRedit: section needs to be removed
- FIGURE CALLOUTS: missing callouts for Fig. 1F and 2C-G
- Checklist: in, but general info table not completed
- DATASET EV LEGENDS: Appendix Table S2 and S5 should be renamed to Dataset EV1-EV2 with the corresponding callouts; legends should be removed from Appendix file, and uploaded as a separate tab in each Excel file
- APPENDIX 1 FILE WITH ToC: Appendix file needs to be in PDF format; Appendix Table Legends should be removed from ToC and Appendix PDF;
- SOURCE DATA: For EV and/or appendix figures, ZIP together all source data.
- Synopsis: Papers published in The EMBO Journal are accompanied online by a 'Synopsis' to enhance discoverability of the manuscript. It consists of A) a short (1-2 sentences) summary of the findings and their significance, B) 3-4 bullet points highlighting key results and C) a synopsis image that is 550x300-600 pixels large (width x height, jpeg or png format). You can either show a model or key data in the synopsis image. Please note that the image size is rather small and that text needs to be readable at the final size. Please send us this information together with the revised manuscript.
- The re-use of western blot between Figure 2F and Figure 6D has to be explicitly mentioned in the figure legends.
- Please indicate the statistical test used for data analysis in the legend of figure 6a.
- Please note that information related to n is missing in the legend of figure 2d.
- Please note that the measure of center for the error bars needs to be defined in the legends of figures 2e; 5a-c; 6a.
- Please note that the yellow arrows are not defined in the legend of figure 2g. This needs to be rectified.
- Appendix tables S1, S3 and S4 should be renamed to Table EV1-EV3 with the corresponding callouts, and legends should be removed from Appendix file, inserted above the tables

With best regards,

Cornelius Schneider

Cornelius Schneider, PhD
Editor | The EMBO Journal
c.schneider@embojournal.org

- a point-by-point response to the referees' comments, with a detailed description of the changes made (as a word file).
- a word file of the manuscript text.

- individual production quality figure files (one file per figure)

- a complete author checklist, which you can download from our author guidelines

(<https://www.embopress.org/page/journal/14602075/authorguide>).

- Expanded View files (replacing Supplementary Information)

We realize that it is difficult to revise to a specific deadline. In the interest of protecting the conceptual advance provided by the work, we recommend a revision within 3 months (5th Aug 2024). Please discuss the revision progress ahead of this time with the editor if you require more time to complete the revisions. Use the link below to submit your revision:

Referee #1:

The authors have carefully considered the reviewer's comments and provided thoughtful, well-reasoned responses to my queries. Overall, this rigorous and ambitious study offers a protein-protein interaction map of gas vesicle proteins. The study lays the foundation for future mechanistic assembly mechanisms and studies of gas vesicles and may aid in customized rationale design of gas vesicle structural properties.

No major or minor concerns.

non-essential suggestions for improving the study:

Setting the low threshold at 5% signal is a bit arbitrary. I'm wondering if you reference the average and standard deviation of the negative controls in your current data set (e.g., FKBP12 or FRB fusion protein with a GV fusion protein). Then use this number to justify your 5% threshold value for low signal.

Referee #2:

The authors have addressed the relatively minor concerns I raised during the first round of review. They are congratulated on what will be a valuable contribution to the literature.

Referee #3:

The authors have addressed all our initial concerns and suggestions satisfactorily. We think this study will be a valuable resource for the gas vesicle community and we applaud the authors for making their plasmids publicly available in a non-profit repository.

All editorial and formatting issues were resolved by the authors.

Dear Dr. Lu,

I am pleased to inform you that your manuscript has been accepted for publication in the EMBO Journal.

Yours sincerely,

Cornelius Schneider, PhD
Editor
The EMBO Journal
c.schneider@embojournal.org
